# Long-term analysis of cryoseismic events and associated ground thermal stress in Adventdalen, Svalbard

Rowan Romeyn[1], Alfred Hanssen[1], Andreas Köhler[1,2]

[1]Department of Geosciences, University of Tromsø – The Arctic University of Norway, 9037 Tromsø, Norway
[2]NORSAR, Gunnar Randers vei 15, 2007 Kjeller, Norway

*Correspondence to*: Rowan Romeyn (rowan.romeyn@uit.no)

**Abstract.** The small-aperture Spitsbergen seismic array (SPITS) has been in continuous operation at Janssonhaugen for decades. The high Artic location in the Svalbard archipelago makes SPITS an ideal laboratory for the study of cryoseisms, a nontectonic class of seismic events caused by freeze processes in ice, ice-soil and ice-rock materials. We extracted a catalogue of >100 000 events from the nearly continuous observation period between 2004 and 2021, characterized by short duration ground shaking of just a few seconds. This catalogue contains two main subclasses where one subclass is related to underground coal mining activities and the other is inferred to be dominated by frost quakes resulting from thermal contraction cracking of ice wedges and crack filling vein-ice. This inference is supported by the correspondence between peaks in observed seismicity with peaks in modelled ground thermal stress, based on a Maxwellian thermo-viscoelastic model constrained by borehole observations of ground temperature. The inferred frost quakes appear to be dominated by surface wave energy and SPITS proximal source positions, with three main areas that are associated with dynamic geomorphological features, i.e., erosional scarps and a frozen debris/solifluction lobe. Seismic stations providing year-round, high temporal resolution measurements of ground motion may be highly complementary to satellite remote sensing methods, such as InSAR, for studying the dynamics of periglacial environments. The long-term observational record presented in this study, containing tens of thousands of cryoseismic events, in combination with a detailed record of borehole ground temperature observations, provides a unique insight into the spatiotemporal patterns of cryoseisms. The observed patterns may guide the development of models that can be used to understand future changes to cryoseismicity based on projected temperatures.

## 1    Introduction

Cryoseisms are a nontectonic class of seismic events caused by freeze processes in ice, ice-soil and ice-rock materials (Lacroix, 1980). For example, the buildup of thermal stress in frozen soils during intense periods of cooling can lead to tensional fracturing and explosive stress release (Barosh, 2000; Battaglia et al., 2016). Since this stress release triggers the propagation of seismic waves, these events are sometimes referred to as frost quakes (e.g., Okkonen et al., 2020). Observations of frost quakes are limited because seismic amplitudes decay rapidly with distance from the point of rupture, but they have been felt at distances of several hundred meters to several kilometers and are usually accompanied by cracking or booming noises,

resembling falling trees, gunshots or underground thunder (Leung et al., 2017; Nikonov, 2010). Frost quakes have typically been observed in association with rapid air and ground cooling, in the absence of an insulating snow layer and where sufficient moisture is present for ice to form (Barosh, 2000; Battaglia et al., 2016; Matsuoka et al., 2018; Nikonov, 2010). Ice can initiate ground cracking by different mechanisms including thermal contraction and tensile failure (e.g., Lachenbruch, 1962) or by the growth of segregation ice driven by the capillary migration of water to a freezing front (Peppin and Style, 2013; Walder and Hallet, 1985). The resulting fractures may potentially cause damage to buildings and other infrastructure in cold regions (Okkonen et al., 2020). Frost damage can be understood as a combination of slow creep (frost heave, e.g., Rempel (2010)) and rapid elastic (frost quake) deformation of frozen ground and causes damage to roads requiring billions of dollars annually to repair in the United States alone (DiMillio, 1999).

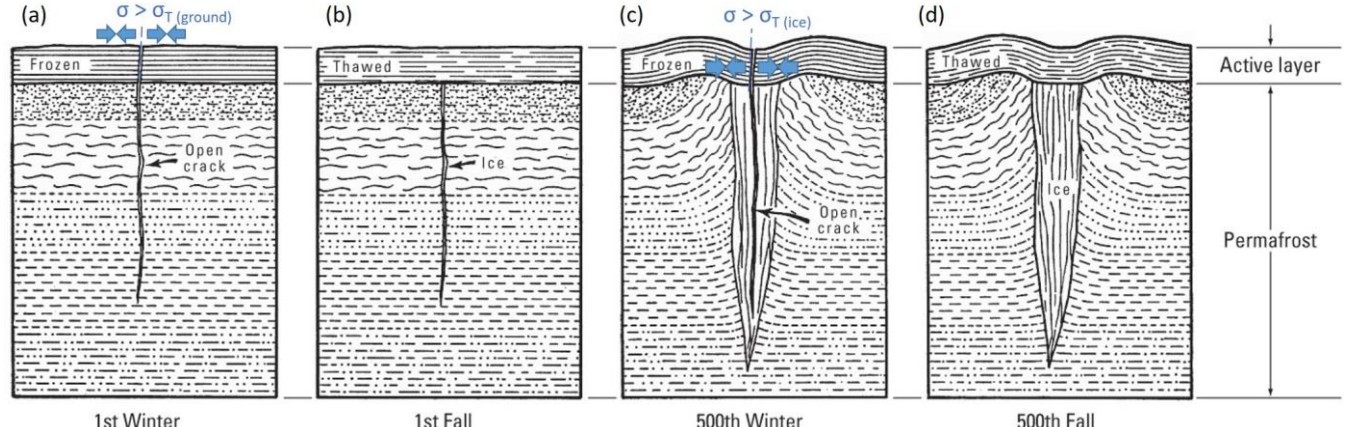

**Figure 1 – modified after Lachenbruch (1962) model of ice wedge formation. (a) Thermal contraction of frozen active layer initiates a tensional crack that penetrates into permafrost when the stress, σ, exceeds the tensile strength of frozen ground, $\sigma_{T(ground)}$, (b) meltwater infiltrates and refreezes during the thaw season, (c) ice with lower tensile strength, $\sigma_{T(ice)}$, than surrounding ground forms a plane of weakness and cracks repeatedly over many years, (d) the crack-infilling-freeze cycle causes ice wedge growth in the permafrost and ground surface deformation that organizes into ice wedge polygons in 2D plan view (e.g., Plug and Werner, 2002).**

In some cases, frost quakes are involved in forming various geomorphic features on the surface. For example, ice-wedge or sand-wedge polygons are a widely observed geomorphic feature in the periglacial environment (e.g., Black, 1976; Matsuoka et al., 2004). These wedges form when water that infiltrates and freezes to ice, or wind transported sand grains, hold open an initial thermal contraction crack that subsequently becomes an enduring plane of structural weakness (Lachenbruch, 1962; Mackay, 1984; Matsuoka et al., 2004; Plug and Werner, 2002; Sørbel and Tolgensbakk, 2002). As shown in Figure 1, repeated cracking, infilling and refreezing along these planes of weakness causes the ice wedges to grow laterally in permafrost. This forces the displaced ground upwards and results in a series of troughs/ridges in a polygonal arrangement that are one of the most recognizable landforms in permafrost environments (Christiansen et al., 2016; Lachenbruch, 1962; Matsuoka et al., 2018; Plug and Werner, 2002). For this reason, sand-wedge polygons, that were formed at sea level on the paleo-equator during

late Neoproterozoic glacial episodes, have been used as evidence supporting the snowball Earth hypothesis (Maloof et al., 2002). Small-scale polygonal features observed on the surface of Mars have also been inferred to result from thermal contraction cracking (Mellon, 1997).


Frost cracking driven by segregation ice growth is also an important agent of bedrock erosion in cold mountainous areas, where rockfall, active screes and high headwall erosion rates are observed in areas where frost action is most intense (Hales and Roering, 2009; Hales and Roering, 2007; Scherler, 2014). An important mode of crack growth in water permeable bedrock is the migration of water to form segregation ice bodies (Hallet et al., 1991; Murton et al., 2006; Walder and Hallet, 1985) that
is similar to the mechanism by which ice lenses develop in freezing soil (Peppin and Style, 2013). On slopes, segregation ice growth, frost heaving and creep leads to the development of solifluction lobes and sheets  (Cable et al., 2018; Matsuoka, 2001). Solifluction is broadly defined as the slow mass wasting resulting from freeze-thaw action in fine-textured soils (French, 2017; Matsuoka, 2001) and occurs due to the asymmetry between frost heaving perpendicular with the sloped ground surface and vertical subsidence upon thawing under the force of gravity.


Spatiotemporal records with high resolution are essential to better understand the processes mentioned above and predict their evolution, for example, in a warming climate. Interferometric synthetic aperture radar (InSAR) is a satellite remote sensing technique capable of resolving down to millimeter scale ground surface displacements at a spatial scale of tens of meters (Hanssen, 2001; Rosen et al., 2000). Consequently, InSAR has been used to resolve seasonal patterns of frost heave and thaw
subsidence (Chen et al., 2020; Rouyet et al., 2021; Wahr et al., 2008). The broad spatial coverage and high spatial resolution of InSAR has also been used to show that the spatial patterns of ground displacement are related to specific geomorphological units, consisting of specific frost prone sediment types and periglacial landforms (Liu et al., 2018; Rouyet et al., 2019). On the other hand, the temporal resolution is determined by the repeat cycle of the satellite (~6 days for Sentinel-1), which means that it is not possible to distinguish between slow frost creep and rapid elastic deformation (frost quake) processes. In addition,
snow cover causes a loss of radar coherence between repeat satellite passes that limits InSAR to snow free areas and seasons. As a result, InSAR is more suited to studying thaw subsidence than frost heave in areas like Svalbard that are snow covered for the majority of the freezing season (e.g., Rouyet et al., 2019). Seismic stations providing year-round, high temporal resolution and highly sensitive measurements of ground motion, at the expense of more limited spatial source resolution and coverage, may be highly complementary to InSAR methods for studying the dynamics of periglacial environments.


The Arctic archipelago of Svalbard is an ideal place to study these processes, given its rapidly warming climate and strong multi-decadal focus on environmental research that has yielded some unique observational datasets. This study was motivated by the intermittent observation of clusters of events recorded by the small-aperture Spitsbergen seismic array (SPITS), located on the main island of the archipelago. These seismic events had durations of just a few seconds (indicative of a nearby source)
and peak amplitudes significantly above the background noise level. For comparison, regional tectonic earthquakes are

typically associated with >30 s ground shaking duration since the different velocities of P, S and surface waves causes the wavefield to spread out (dispersion) as the source distance increases. SPITS has previously been used to study a class of cryoseismic signals associated with iceberg calving at the termini of grounded tidewater glaciers at local to near-regional distances of up to 200 km (Köhler et al., 2015). These calving related seismic signals occurred most frequently during the melt season and the ground motion lasts ~15-20 s. Signals with intermediate durations have also been observed at SPITS, originating from nearby mountain glaciers (Albaric et al., 2021) and coal mining operations (Gibbons and Ringdal, 2006).

Based on previous work further down-valley in Adventdalen, e.g., Matsuoka et al. (2018) and Romeyn et al. (2021), we hypothesized that the short duration events at SPITS might be dominated by frost quakes initiated by thermal contraction cracking in the local vicinity of the array. The aim of this study was to test this hypothesis by analyzing the spatial and temporal occurrence of these events. We focus on the thermal contraction cracking mechanism because it is consistent with the association of transient ground acceleration events with rapid cooling episodes and cold winter temperatures reported by Matsuoka et al. (2018) and consistent with previous descriptions of frost quakes (Barosh, 2000; Battaglia et al., 2016; Nikonov, 2010; Okkonen et al., 2020). This study is highly complementary to the previous work of Romeyn et al. (2021). While the spatial and temporal wavefield sampling of the SPITS array is much coarser than the temporary geophone array they deployed, a much longer and nearly continuous record is available. This allows a more rigorous investigation of the temporal correlation of these events with ground cooling and thermal stress accumulation.

## 2    Study area and data

The small-aperture Spitsbergen seismic array (SPITS) is located on Janssonhaugen, in the Adventdalen valley on the island of Spitsbergen, part of the high-Arctic Svalbard archipelago (Figure 2). The SPITS array has been in operation since 1992, maintained by the research foundation NORSAR (Schweitzer et al., 2021). At present, it consists of 9 CMG-3T seismometers with an aperture of 1 km and interstation distances >250 m (e.g., Gibbons et al., 2011; Köhler et al., 2015) installed in shallow boreholes below the permafrost active layer. The standard frequency response of the CMG-3T seismometer is flat (within -3dB) for the range from 0.0083 to 50 Hz. We chose to limit the present study to the period following August 2004 when the SPITS array was upgraded to a full broadband array with an increase in sampling rate from 40 to 80 Hz for all seismometers (Schweitzer et al., 2021). The waveform data following the upgrade is of high quality and well suited to source localization using matched field processing. Waveform data for the SPITS array were retrieved from the European Integrated Data Archive (EIDA), maintained by the University of Bergen and NORSAR.

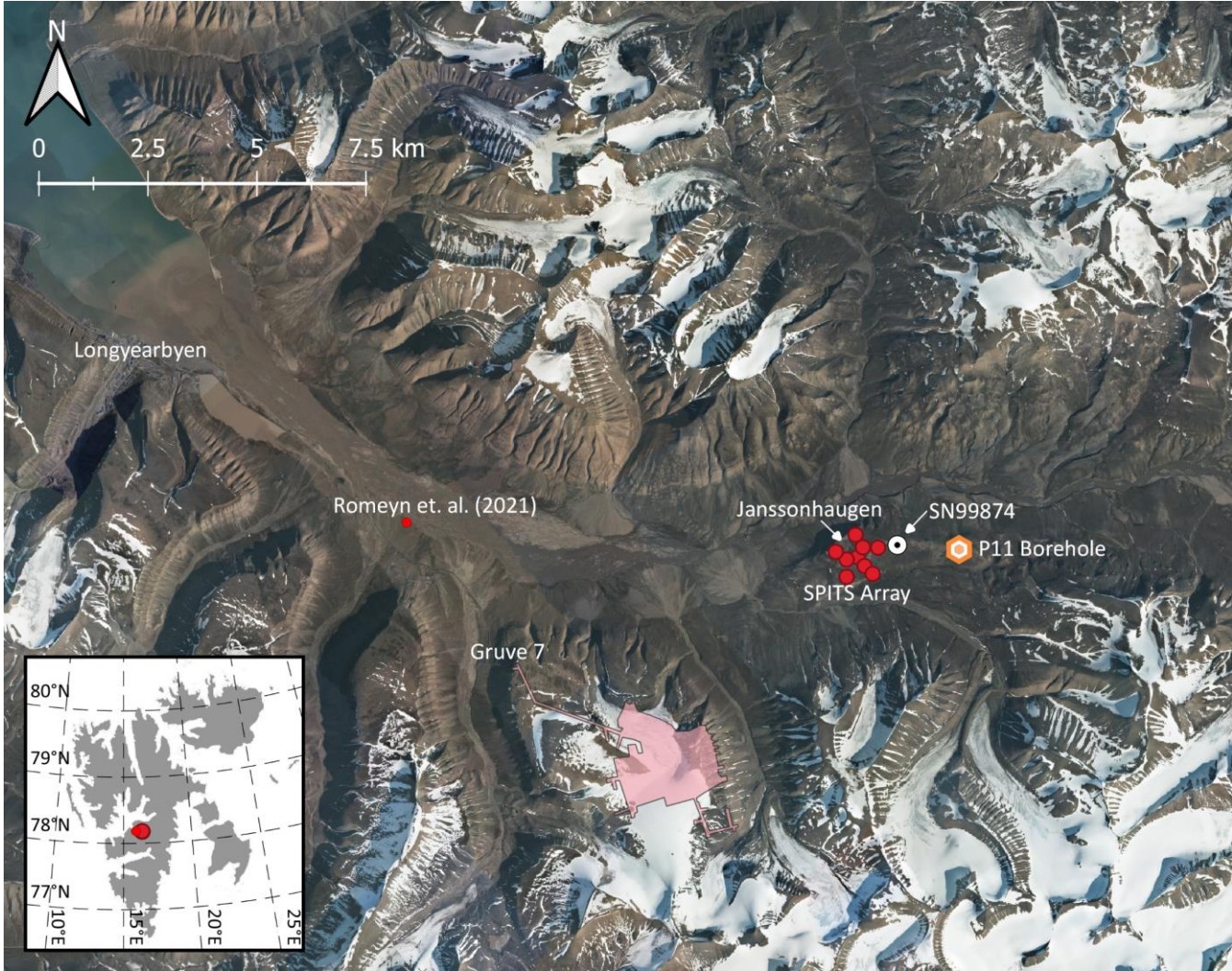

**Figure 2 – Overview of the location of the SPITS array located on Janssonhaugen in Adventdalen, Svalbard, up-valley from the settlement of Longyearbyen and the temporary seismic array of Romeyn et al. (2021). The P11 temperature logging borehole and the underground mining areas and tunnels of the operational coal mine, Gruve 7, are also shown. The met.no weather station "Janssonhaugen Vest" is marked by its station number, SN99874. Orthophoto © Norwegian Polar Institute (npolar.no).**

Janssonhaugen is a bedrock remnant located in the middle of the Adventdalen valley, with a ~0.2-0.3 m thick weathered sediment crust overlying homogeneous sand-/siltstone bedrock (density 2280 kg.m$^{-3}$, porosity 20-25 %, >96.5 % $SiO_2$) corresponding to the Ullaberget Member of Lower Cretaceous Rurikfjellet Formation (Dypvik et al., 1991; Isaksen et al., 2001). The bedrock at Janssonhaugen has been drilled to a depth of 102 m and the permafrost zone is estimated to extend down to 220 m, based on downward extrapolation of the measured temperature gradient (Isaksen et al., 2001). Despite annual precipitation of 300-500 mm/yr., the snow cover on Janssonhaugen is typically thin or completely absent due to the scouring effect of the prevailing winds (Isaksen et al., 2001). The surface topography is generally flat but loose surface material is sorted

into polygons (Isaksen et al., 2001), due to active layer cryoturbation and/or ice wedge formation. The homogeneous geology at Janssonhaugen, it's flat topography, limited snow cover, relatively large distance to glaciers, rivers, ocean, human activity such as coal mining and position well above the Holocene marine limit make it a good location to study permafrost processes (Isaksen et al., 2000).

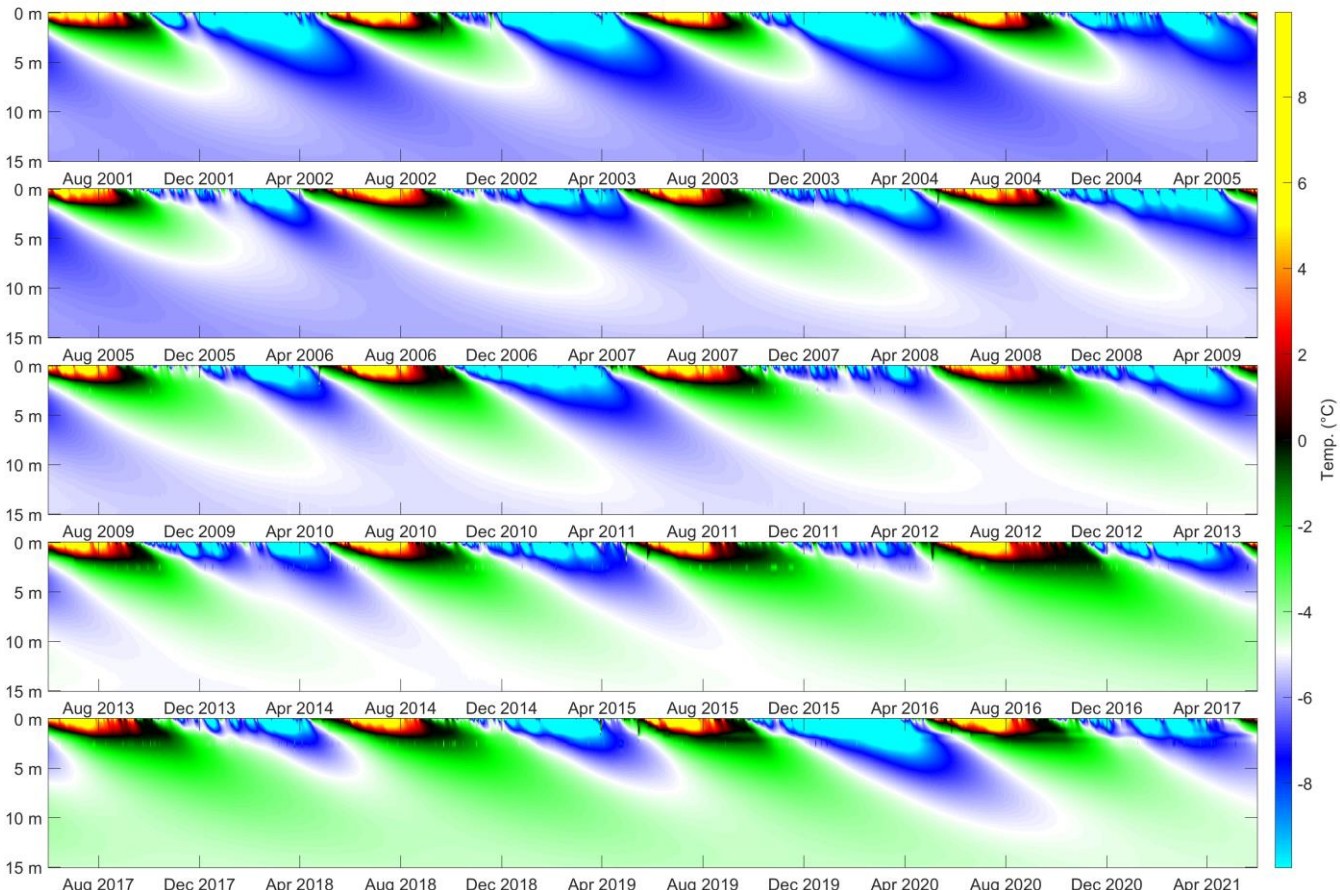

**Figure 3 – Illustration of spatiotemporal borehole temperature recorded at the PACE P11 borehole on Janssonhaugen. A long-term warming trend is observed in the permafrost below the ~2 m thick active layer that is subject to seasonal freeze-thaw. The continuous timeline is split across multiple figure panels. Temperature-depth profile was interpolated to regular 10 cm intervals using a spline interpolant.**

At this location, the polar night, where no shortwave solar radiation is received at the ground surface, lasts from around 1-Oct to 28-Feb each year, but is nonetheless the season associated with the largest diurnal temperature range (Przybylak et al., 2014). This is because intense winter storms promote the advection of warmth to the region, driven 95% by atmospheric circulation and 5% by oceanic circulation (Bednorz, 2011). Snow cover and latent heat fluxes also contribute to a complex surface energy budget (Westermann et al., 2009). As a result of the complex interplay of processes driving temperature variation in this high-Arctic location, temperature monitoring boreholes installed at Janssonhaugen are critical to our ability to

accurately model the subsurface buildup of thermal stress. Thermistors installed in the GTN-P (Global Terrestrial Network for Permafrost) boreholes P10 (102 m deep) and P11 (15 m deep) provide a continuous record of the subsurface temperature field at Janssonhaugen, with a sampling interval of 6 hours extending from April 1999 to the present (Isaksen et al., 2001). We focus on the P11 borehole (see Figure 2) that was drilled 13 m horizontally offset from P10, gives the most detailed record of the near-surface temperature field and is least disturbed by installations at the ground surface (Isaksen et al., 2000). The temperature field measured by the P11 borehole is illustrated in Figure 3. The ~2 m thick active layer (Christiansen et al., 2020) is sampled by thermistors at 0.2, 0.4, 0.8, 1.2, 1.6 and 2 m and there is significant inter-annual variability in the magnitude of summer warming and winter cooling. The upper part of the permafrost at the P11 borehole location is sampled by thermistors installed at 2.5, 3, 3.5, 4, 5, 7, 10, 13 and 15 m. These measurements record a long-term warming trend in the permafrost beneath the active layer (see Figure 3). Furthermore, the Janssonhaugen Vest weather station (see Figure 2), which was installed in September 2019, includes hourly sampled records of air temperature and ground temperature at 0.1 m depth. It therefore provides a basis to compare depth and temporal sampling effects against the longer duration, more coarsely sampled P11 borehole record.

## 3    Methods

### 3.1    STA/LTA detection of short duration seismic events

Events are detected based on anomalous values of short-time-averaged (STA) amplitude divided by long-time-averaged amplitude (LTA), i.e., the classic STA/LTA approach widely used in seismology (e.g., Allen, 1982; Trnkoczy, 2009). The purpose of the event detector is to make an initial, coarse, automatic identification of short duration seismic signals, which should be distinguished from both background noise and longer duration local and regional seismic events that may be high amplitude. The raw data from each seismometer was first de-trended, corrected according to the calibrated instrument sensitivity and bandpass filtered to the range 2.5-20 Hz using a delay-compensated minimum phase filter with a stopband attenuation of 60 dB (see Figure 4a). This passband was selected to eliminate high-frequency random noise which can lead to spurious spikes in STA/LTA.

The STA window length should be comparable to the target signal duration, while the LTA represents the background noise level. When the STA/LTA ratio exceeds a given threshold, an event is triggered. In our implementation, the STA is given by the one-second moving-average smoothed trace envelope for each seismogram. The LTA is the STA further smoothed according to a 20 second period moving average. Since we have an array of stations, we represent the array-STA by taking the 80[th] percentile station-STA across all stations at a given time sample. If we had chosen the maximum station STA, we may detect arbitrary noise spikes with large amplitudes registered on a single seismometer. If the mean or median station STA were chosen, we would preferentially detect larger regional events with more consistent amplitudes across the array and suppress smaller local events with high amplitudes limited to a small subset of seismometers.

The STA/LTA threshold was set to 10. Furthermore, after a trigger, no new events are declared within 5 seconds after the STA/LTA threshold was exceeded to avoid detecting the same events multiple times. All processing parameters (STA and LTA lengths, STA/LTA threshold, STA percentile, trigger pause) were found to be appropriate for detection of short-duration signals coherent across the array, while avoiding false triggers (noise bursts at single stations), by visually inspecting test

periods. We further reject epochs with LTA more than 2.5 times the 2-hour mean of the LTA in order to filter out large regional earthquakes. An example of event detection based on STA/LTA peaks is shown in Figure 4b, which illustrates that short duration events are selected while a longer duration high-amplitude event is not detected as intended.

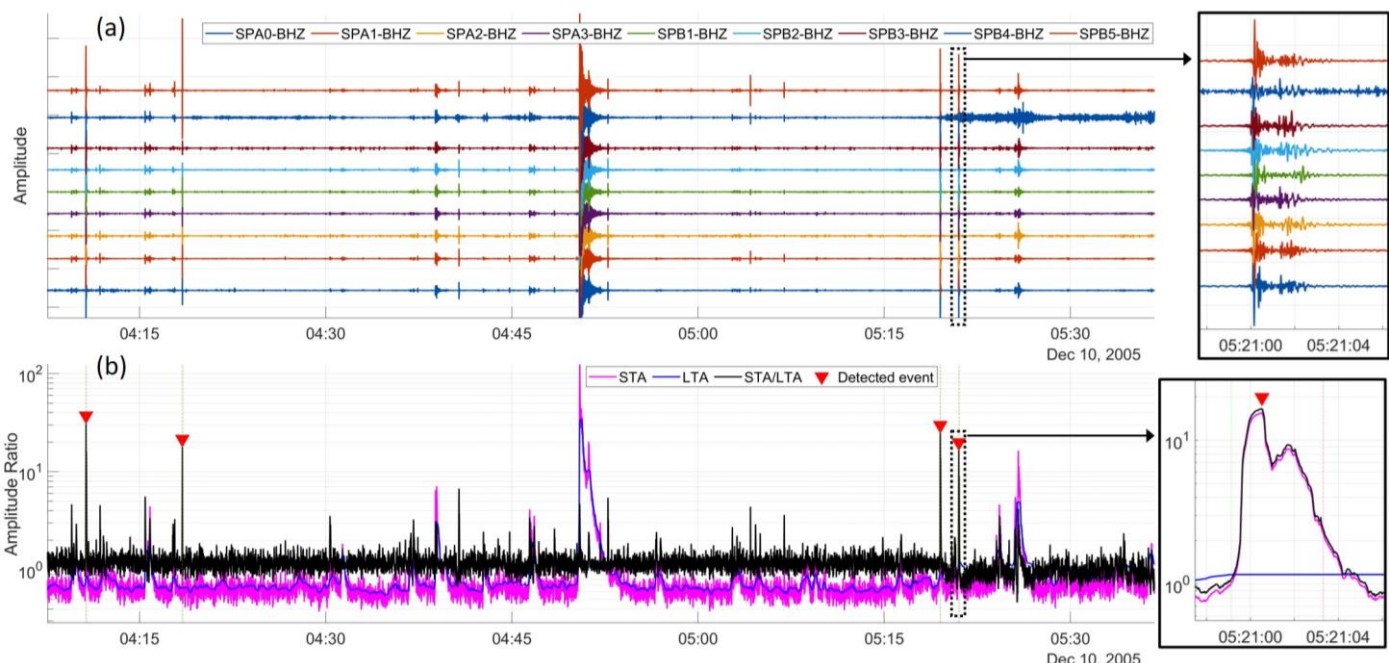

**Figure 4 – Example of (a) timeseries of vertical component ground motion and (b) event detection using the STA/LTA detector. Short duration events with sufficient amplitude and array coherence are selected while longer events such as the high amplitude example at 04:50 are ignored. Inset boxes show detailed views for a specific detection.**

### 3.2 Source localization by coherent MFP

Matched-field processing (MFP) is an established technique for localizing the position of seismic sources recorded by passive

seismic arrays (Chmiel et al., 2016; Cros et al., 2011; Harley and Moura, 2014; Sergeant et al., 2020; Walter et al., 2015). MFP proceeds via an evaluation of the similarity between the wavefield recorded at a receiver array and a series of predicted wavefields calculated for a grid of test source locations and a theoretical model of the source-receiver wave propagation. The MFP coherence is estimated by comparing the recorded data vector with a "replica" vector, which is a model representation of wave propagation within the medium. We assume a simple homogeneous medium with amplitude decay according to

spherical divergence where the replica column vector, $\boldsymbol{R}'$ is represented by the theoretical harmonic wave emitted from a test point $p(x, y)$ according to

$$\boldsymbol{R}'(\omega, \boldsymbol{d}) = \left[ \frac{1}{d_1} e^{i\omega d_1/c(\omega)}, \frac{1}{d_2} e^{i\omega d_2/c(\omega)}, \dots, \frac{1}{d_N} e^{i\omega d_N/c(\omega)} \right]^T / \eta. \tag{1}$$

Here, $i = \sqrt{-1}$, $\omega$ is the angular frequency, $\boldsymbol{d} = [d_1, d_2, \cdots, d_N]^T$ is an $N \times 1$ vector containing the absolute Euclidean distances between $p(x, y)$ and the $N$ recording stations where superscript $T$ denotes a transpose, $c(\omega)$ is the medium phase velocity at frequency $\omega$ and the vector is normalised to unit length by dividing by the factor $\eta = \sqrt{\sum_{j=1}^{N} (1/d_j^2)}$. The column data vector is given by

$$\boldsymbol{R}(\omega) = [R_1(\omega), R_2(\omega), \dots, R_N(\omega)]^T, \tag{2}$$

where $R_j(\omega)$ is a frequency transform of the $j$-th trace $r_j(t)$ of $N$ traces recording a specific seismic event. We then form the complex-valued $N \times N$ cross-spectral density matrix (CSDM) by

$$\mathbf{K}(\boldsymbol{\omega}) = \boldsymbol{R}(\boldsymbol{\omega}) \boldsymbol{R}^H(\boldsymbol{\omega}), \tag{3}$$

where $(\cdot)^H$ denotes the Hermitian (conjugate transpose) operator. The frequency domain transform was implemented via the chirp z-transform (Rabiner et al., 1969), which provides a convenient means to evaluate the band limited transform. In this study we selected the 5-35 Hz band, sampled at an interval of 1 Hz, in order to reject spatially coherent low-frequency background noise while retaining the shared signal/noise high-frequency band (Appendix D shows that the results are relatively insensitive to the selected frequency band). This allows us to efficiently and compactly represent the CSDM, whose size is a significant factor in the speed of computation, in addition to the number of test source points, $p(x, y)$. Conventionally, the MFP coherence is estimated using the linear Bartlett processor

$$G = \sum_{\omega} \boldsymbol{R}'^H(\boldsymbol{\omega}, \boldsymbol{d}) \mathbf{K}(\boldsymbol{\omega}) \boldsymbol{R}'(\boldsymbol{\omega}, \boldsymbol{d}) \tag{4}$$

which evaluates the inner product between the recorded and predicted wavefields before summing incoherently across frequency. The matrix operations in Eq. (4) are quadratic forms that formally guarantee a non-negative real-valued output.

In this study we implement the coherent MFP scheme developed by Michalopoulou (1998), which is an elegant addition to the conventional approach allowing cross-frequency spatial coherence structures to be exploited to give improved robustness and accuracy. In this scheme, measurement vectors at $L$ discrete frequencies are concatenated to form the $NL \times 1$ measurement super-vector

$$\boldsymbol{\mathcal{R}}(\boldsymbol{\omega}) = [\boldsymbol{R}(\boldsymbol{\omega_1}), \boldsymbol{R}(\boldsymbol{\omega_2}), \cdots, \boldsymbol{R}(\boldsymbol{\omega_L})]^T, \tag{5}$$

where $\boldsymbol{\omega} = [\omega_1, \omega_2, \cdots, \omega_L]^T$ is a frequency vector, and the replica vectors are concatenated to form the $NL \times 1$ replica super-vector

$$\boldsymbol{\mathcal{R}}'(\boldsymbol{\omega}, \boldsymbol{d}) = [\boldsymbol{\mathcal{R}}'(\boldsymbol{\omega_1}, \boldsymbol{d}), \boldsymbol{\mathcal{R}}'(\boldsymbol{\omega_2}, \boldsymbol{d}), \cdots, \boldsymbol{\mathcal{R}}'(\boldsymbol{\omega_L}, \boldsymbol{d})]^T. \tag{6}$$

The super-CSDM $\mathcal{K}(\omega) = \mathcal{R}(\omega)\mathcal{R}^H(\omega)$ is then composed of $NL \times NL$ elements and the generalized MFP coherence is given by

$$\mathcal{G} = \mathcal{R}'^H(\omega, d) \, \mathcal{K}(\omega) \, \mathcal{R}'(\omega, d). \tag{7}$$

The estimated source position, $p(x, y)$, and phase velocity, $c(\omega)$, are those which maximize the coherence measure $\mathcal{G}$. Since some of the seismic events we wish to locate are dominated by surface waves and others are dominated by body waves, we

simply assume that phase velocity is a constant and scan over the range 250-6000 m/s.

## 3.3 Ground thermal stress model

Previous studies such as Lachenbruch (1962), Mellon (1997), Maloof et al. (2002), Schulson and Duval (2009) and Podolskiy et al. (2019) have demonstrated that ice and frozen ground deform elastically on short timescales and viscously on long timescales. Thermal loading due to temperature changes acts as an external driving agent, and the resulting dynamical balance

between the elastic and viscous response governs whether creep or fracture become dominant. Following Mellon (1997), we neglect the layered structure of the ground and model the frozen ground as a simple homogeneous Maxwellian viscoelastic solid augmented with thermal expansion and contraction. This allows us to decompose the total strain tensor $\varepsilon_{ij}$ into three components: an elastic ($\varepsilon_{ij}^e$), a thermal ($\varepsilon_{ij}^T$), and a viscous ($\varepsilon_{ij}^V$) component,

$$\varepsilon_{ij} = \varepsilon_{ij}^e + \varepsilon_{ij}^T + \varepsilon_{ij}^V, \tag{8}$$

where subscripts $ij$ indicate tensor components, $i, j = 1,2,3$, where 1 and 2 denote horizontal components, and 3 is the vertical component. The elastic strain tensor is related to the stress tensor $\sigma_{ij}$ by (e.g., Landau and Lifshitz, 1970)

$$\varepsilon_{ij}^e = \frac{1+\nu}{E} \, \sigma_{ij} - \frac{\nu}{E} \, \sigma_{kk} \, \delta_{ij}, \tag{9}$$

where $\nu$ is Poisson's ratio, $E$ is Young's modulus, $\delta_{ij}$ is the Kronecker delta, and Einstein's summation convention is applied throughout this paper.

The thermal strain tensor is a measure of the change in volume caused by the thermally driven deformation, and it is expressed as (e.g., Landau and Lifshitz, 1970)

$$\varepsilon_{ij}^T = \alpha(T - T_0) \, \delta_{ij}, \tag{10}$$

where $T$ is the temperature, $T_0$ is a reference temperature for the undeformed state, and $\alpha$ is the linear thermal expansion

coefficient. The viscous strain tensor is a complicated topic in itself, and a wide range of phenomenological and heuristic parametric models for the viscous strain rate exist (e.g., Bingham, 1922; Carreau, 1972; Glen, 1955; Herschel, 1926; Saramito, 2007). To encompass this generality, we formulate the viscous strain rate as

$$\frac{\partial \varepsilon_{ij}^V}{\partial t} = \Gamma_N\{s_{ij}\}, \tag{11}$$

where $\Gamma_N\{\cdot\}$ is a nonlinear operator acting on the deviatoric stress $s_{ij} = \sigma_{ij} - \sigma_{kk}/3$. The chosen parametric form of $\Gamma_N\{\cdot\}$ determines how induced stress relaxes in the medium, and both Newtonian and non-Newtonian behavior can be incorporated in this formulation.

Following Mellon (1997), we assume $\sigma = \sigma_{11} = \sigma_{22}$ and $\partial\varepsilon_{11}/\partial t = \partial\varepsilon_{22}/\partial t = 0$. The vertical stress is also neglected, i.e., $\sigma_{33} \approx 0$, corresponding to the assumption that the overburden is negligible in the shallow subsurface (Mellon, 1997). We assume that Poisson's ratio, $\nu = \nu(T)$, Young's modulus, $E = E(T)$, and the coefficient of linear thermal expansion, $\alpha = \alpha(T)$ are all temperature dependent, and thus, implicitly time dependent since $T = T(z,t)$ is the temperature at depth $z$ and time $t$. By direct evaluation of $\partial\varepsilon_{ij}/\partial t$ and collecting terms, we find that the temporal dynamics of horizontal stress $\sigma(z,t)$ in a Maxwellian viscoelastic solid driven by thermal expansion and contraction is governed by the following first-order nonlinear and nonhomogeneous differential equation

$$\frac{\partial\sigma}{\partial t} + \beta(t)\sigma + \Gamma\{\sigma\} = \kappa(t). \tag{12}$$

The time-dependent coefficients are found to be

$$\beta(t) = -\left(\frac{1}{E}\frac{\partial E}{\partial T} + \frac{1}{1-\nu}\frac{\partial\nu}{\partial T}\right)\frac{\partial T}{\partial t}, \tag{13}$$

$$\kappa(t) = -\frac{E}{1-\nu}\left[\alpha + \frac{\partial\alpha}{\partial T}(T - T_0)\right]\frac{\partial T}{\partial t}, \tag{14}$$

and $\Gamma\{\cdot\} = \frac{E}{1-\nu}\Gamma_N\{\cdot\}$. The scientific literature devoted to the rheology of frozen materials favor power-law parametrizations for the viscous term (e.g., Schulson and Duval, 2009). In this paper, we apply the heuristic temperature dependent power-law proposed by Glen (1955) in the form used by Mellon (1997) and Maloof et al. (2002):

$$\Gamma\{\sigma\} = \frac{E}{1-\nu}A_0\left|\frac{\sigma}{2}\right|^n \text{sign}(\sigma)\exp(-Q/RT). \tag{15}$$

In Glen's flow law, $R$ is the universal gas constant, and $A_0$, $Q$, and $n$ are empirical parameters that need to be chosen. The temperature-dependent Arrhenius exponential term in this particular choice of $\Gamma\{\cdot\}$ is included to model the increasing ductility as temperature increases (e.g., Glen, 1955). To sum up, the first two terms on the left-hand side in Eq. (12) are connected to the elastic response of the solid, the third term models viscous relaxation, and the right-hand side is the thermal driving term. In order to solve Eq. (12) for $\sigma(z,t)$, we specify the initial condition $\sigma_0(z) = \sigma(z, t=0) = 0$.

If we assume $\partial\nu/\partial T = 0$, Eq. (12) reduces to the model proposed by Mellon (1997). By contrast, if we assume $\partial\nu/\partial T = \partial E/\partial T = \partial\alpha/\partial T = 0$, Eq. (12) reduces to the model proposed by Podolskiy et al. (2019). Finally, if we assume $\partial\nu/\partial T = \partial E/\partial T = \partial\alpha/\partial T = A_0 = 0$, Eq. (12) reduces to the Timoshenko and Goodier (1951) model for thermal stress as applied by Okkonen et al. (2020), excluding the boundary correction terms that represent compressive stresses at the free surfaces of the finite thickness plate assumed by the latter authors.

We solved Eq. (12) numerically to obtain a time series of the resulting horizontal stress $\sigma(z, t)$, at depth $z$, using the standard Matlab solver ode45, based on the well-known fifth-order Runge-Kutta method (Dormand and Prince, 1980). The crucial input to the forward model is the measured temperature time series $T(z, t)$ at depth $z$. Notably, most previous studies of subsurface thermal stress have relied on thermal conduction models in order to infer subsurface temperature variation from measurements of air temperature. A significant novelty of this study is that the ground temperature profile, $T(z, t)$, at Janssonhaugen has been

logged by a series of thermistors installed in the 15 m deep P11 borehole at 6 hr intervals since April 1999. The set of physical parameters that we assume is described in Table 1 and give a homogenised representation of frozen regolith and soft sandstone interspersed with veins of ice that form planes of structural weakness.

**Table 1 – Physical parameters used in thermal stress model**

| Parameter | Symbol, Unit | Value or Equation | Note or reference |
|---|---|---|---|
| $T$ at zero stress state | $T_0, {}^\circ C$ | 0 | Initial reference temperature |
| Poisson's ratio | $\nu, -$ | $\begin{cases} 0.3 & T \geq 0 \\ 0.008T + 0.3 & -10 \leq T < 0 \\ 0.00067T + 0.23 & T < -10 \end{cases}$ | Generalized from Hu et al. (2013) alluvium, Istomin and Nazarov (2019) and Zhankui et al. (1998) |
| Young's modulus | $E, GPa$ | $\begin{cases} 0.7 & T \geq 0 \\ -0.73T + 0.7 & -10 \leq T < 0 \\ -0.047T + 7.5 & T < -10 \end{cases}$ | Generalized from Draebing and Krautblatter (2012); Timur (1968); Weeks and Assur (1967); Wu et al. (2017) small-strain dynamic elastic modulus of ice, permafrost and frozen rock measured by seismic waves |
| Linear thermal expansion coefficient | $\alpha, {}^\circ C^{-1}$ | $\begin{cases} 10^{-6}\begin{pmatrix} -0.000237T^3 \\ +0.00885T^2 \\ -0.1852T + 52.52 \end{pmatrix} & T \leq 0 \\ 10^{-6}\begin{pmatrix} -0.0621T^2 \\ +5.78T - 22.3 \end{pmatrix} & T > 0 \end{cases}$ | Butkovich (1959) gives $3^{rd}$ order polynomial for ice and a $2^{nd}$ order polynomial was fitted to Kell (1967) water thermal expansion data |
| Viscous pre-factor | $A_0, s^{-1}Pa^{-n}$ | $1 \times 10^{-9}$ | Glen's non-Newtonian power law viscous flow (Behn et al., 2021; Glen, 1955; Weertman, 1983) |
| Viscous activation energy | $Q, J\,mol^{-1}$ | $1.34 \times 10^5$ | |
| Viscous exponent | $n, -$ | 3.2 | |
| Gas constant | $R, J\,mol^{-1}K^{-1}$ | 8.314 | $T$ converted to Kelvin when evaluating $RT$ |
| Tensile strength | $\sigma_T, MPa$ | 1.0 (range from 0.8-1.3 given in reference) | Currier and Schulson (1982), varies according to grain size for randomly oriented polycrystalline ice (finer grained ice stronger), insensitive to temperature (Petrovic, 2003). |

### 3.3.1 Fracture model

Our aim is to model simple thermal tensional cracking of existing ice wedges or crack filling vein-ice and not the initiation or propagation of new cracks into previously undamaged ground, where pore scale fluid migration and stress localisation at crack tips become important (e.g., Walder and Hallet, 1985). To this end, we apply a simplistic model of fracturing by considering the modelled thermal stress as the pre-fracture stress. When the pre-fracture stress exceeds the tensile strength of ice (see Table 1), a frost quake is assumed to occur and dissipate stress corresponding to 100% of the tensile strength. We keep a tally of the number of frost quakes over time and assume that multiple frost quakes occur if the tensile strength is exceeded by an integer factor greater than one. In reality, a fraction of the stress would be redistributed elastically rather than completely lost to friction as in this simple model. Since ground temperature is a measured quantity, we do not account for warming by frictional heat as a result of frost quake movement.

## 4 Results and discussion

A total of 137,532 short duration seismic events were detected by our STA/LTA detector between July 2004 and June 2021. In order to improve the precision of the source localisation by coherent MFP, only events recorded by at least five seismometers (max four inoperative stations) were located, for a total of 137,456 located short duration events. The estimated source locations were subsequently used to identify subclasses of events, as detailed in the following section.

### 4.1 Subclasses of short duration seismic events

We find that there are two main sub-classes of short duration events recorded by the SPITS array. Event class I is characterised by significant amplitude variation and arrival time differences across the array seismometers, as illustrated in Figure 5a/b. Using coherent MFP to infer the source positions of these events, shows that they occur in relatively close proximity and are clustered inside a circle with radius of ~1500 m centred over the array (Figure 5d and 5e). By contrast, Event class II is characterised by similar amplitudes across the array elements and smaller relative arrival time differences, as illustrated in Figure 5c. Using coherent MFP, we find that these events are associated with more distal inferred source positions (Figure 5f) that also have a consistent azimuth. Figure 5 also illustrates the property that coherent MFP decreases source localisation ambiguity for arrays that sample the spatial domain coarsely for a given range of observed wavelengths when compared to the incoherent scheme (consistent with Michalopoulou, 1998). We also observed that the well calibrated amplitude responses of the SPITS seismometers gives good constraint on source range under the assumed model of amplitude attenuation due to geometrical spreading (Figure 5f). In Figure 5i, we see that incoherent averaging (Eq. 4) has enhanced a sidelobe and produced an incorrect source position that is not consistent with the relative arrival times observed in Figure 5c (earliest arrival at station 7 and latest arrivals with weakest amplitudes at stations 5 and 9).

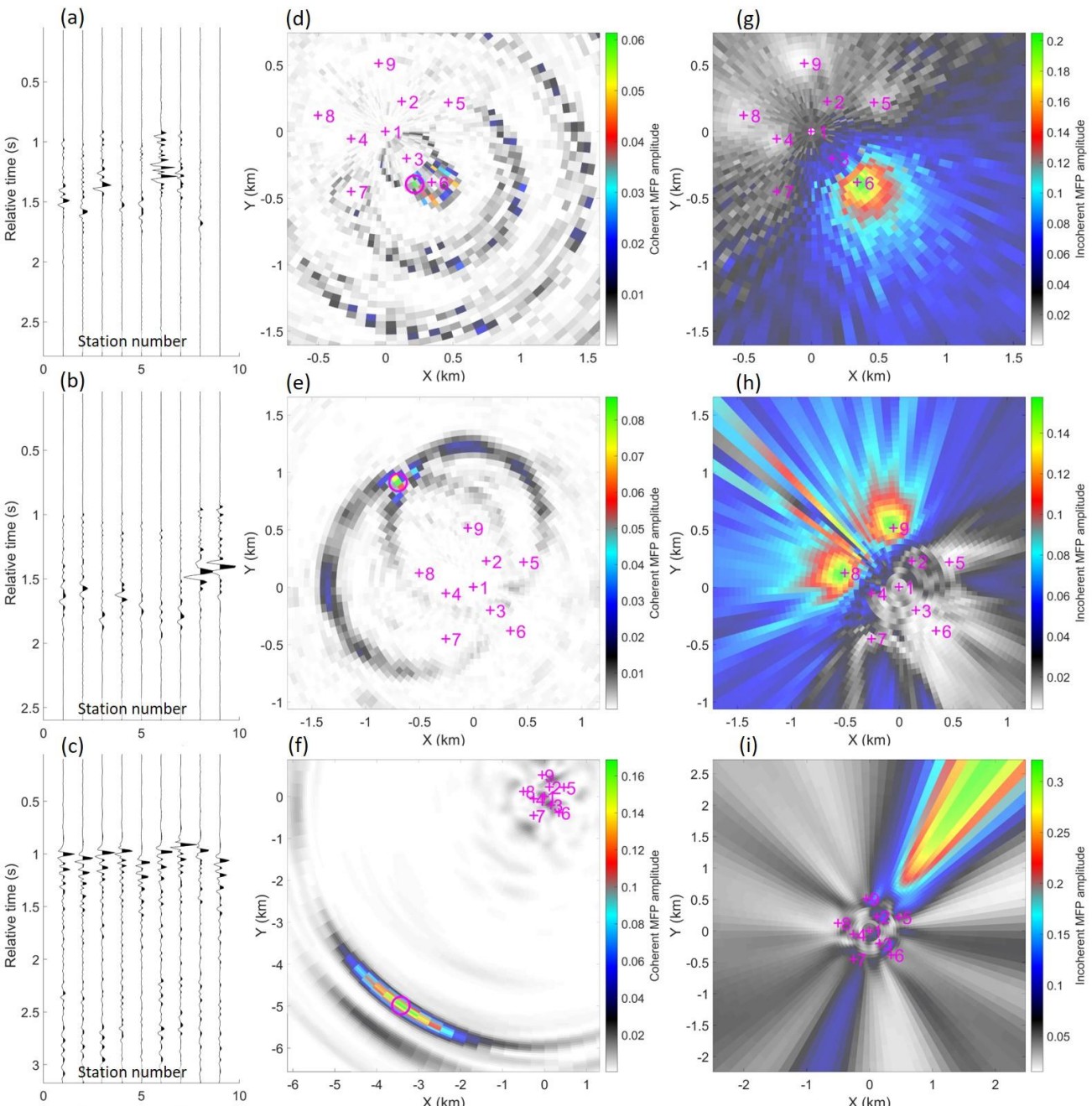

**Figure 5 –** **(a), (b) Examples of Class I events with significant amplitude variation across the array and relatively close source position inferred by (d), (e) coherent MFP, Eq. (7). (c) Example of a Class II event with little amplitude variation across the array and a more distal (f) coherent MFP inferred source position. (g), (h) & (i) show corresponding incoherent MFP results, Eq. (4), demonstrating the improvement gained by coherent MFP. Station numbers in (a), (b) & (c) correspond to the labels annotated on the MFP panels where the seismometer locations are marked with magenta crosses.**

The mean MFP inferred propagation velocities for Class I events was 1150 m/s with a standard deviation of 1100 m/s, implying
a relatively shallow propagation path. The large standard deviation may indicate the surface waves are dispersive with different
frequencies propagating at different phase velocities. However, the signal for a given event was dominated by a relatively
narrow band of frequencies rarely exceeding 5 Hz bandwidth (see Appendix C), so the distinctive highly dispersive waveforms
that one would expect for a wideband dispersive signal were not observed (Figure 5). By contrast, the mean MFP inferred
propagation velocity for Class II events was 5750 m/s with a standard deviation of 400 m/s, indicating that this event class is
dominated by P-wave energy. Three-component seismograms also support the interpretation that event Class I and II are
dominated by surface waves and P-waves respectively (see Appendix C).

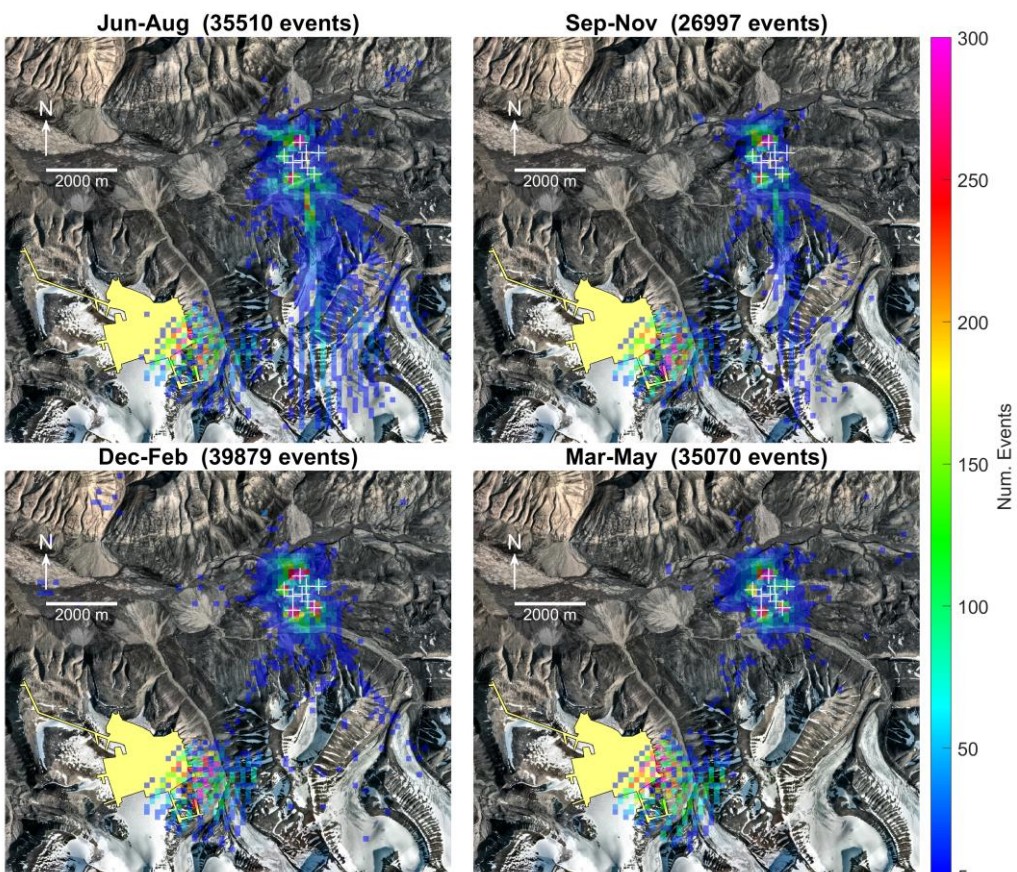

**Figure 6 – Spatially binned (150×150 m) distribution of coherent MFP inferred seismic source positions plotted by season for events recorded between August 2004 and July 2021. Positions of SPITS seismometers are indicated by white crosses. The distal event cluster corresponds with the location of underground mining operations at Gruve 7 (yellow polygon). Orthophoto © Norwegian Polar Institute (npolar.no).**

Mapping the inferred source positions, it is clear that Class II events are spatially coincident with the underground mining
areas of Gruve 7, as illustrated in Figure 6. Furthermore, we observe that Class II events occur frequently during all seasons

(see Figure 6). We infer that Event class II is a result of mining operations and human activity in the underground coal mine, Gruve 7. This is also supported by the dominant P wave of the Class II signals which indicates an explosive source. These events are essentially unwanted noise from the environmental seismology perspective, although they do give a useful indication of the localisation performance of the coherent MFP algorithm as applied in this study. The accuracy of the inferred source positions is largely a function of data versus model phase velocity mismatch and amplitude attenuation that deviates from the

model. The error in the estimated source position will then be proportional to the amplitude spectrum weighted model/data phase velocity mismatch integrated over the signal bandwidth, convolved with the deviation from a purely geometrical spreading model of amplitude attenuation. It is very difficult to quantify this deviation *a priori* so the best way to assess MFP accuracy is by locating known test sources. A set of test sources covering the entire study domain would have been ideal, but the fact that the seismic events associated with mining activities at Gruve 7 show reasonable spatial correspondence with the

location of the underground mine workings is encouraging and indicates that the model assumptions give a reasonable approximation of reality.

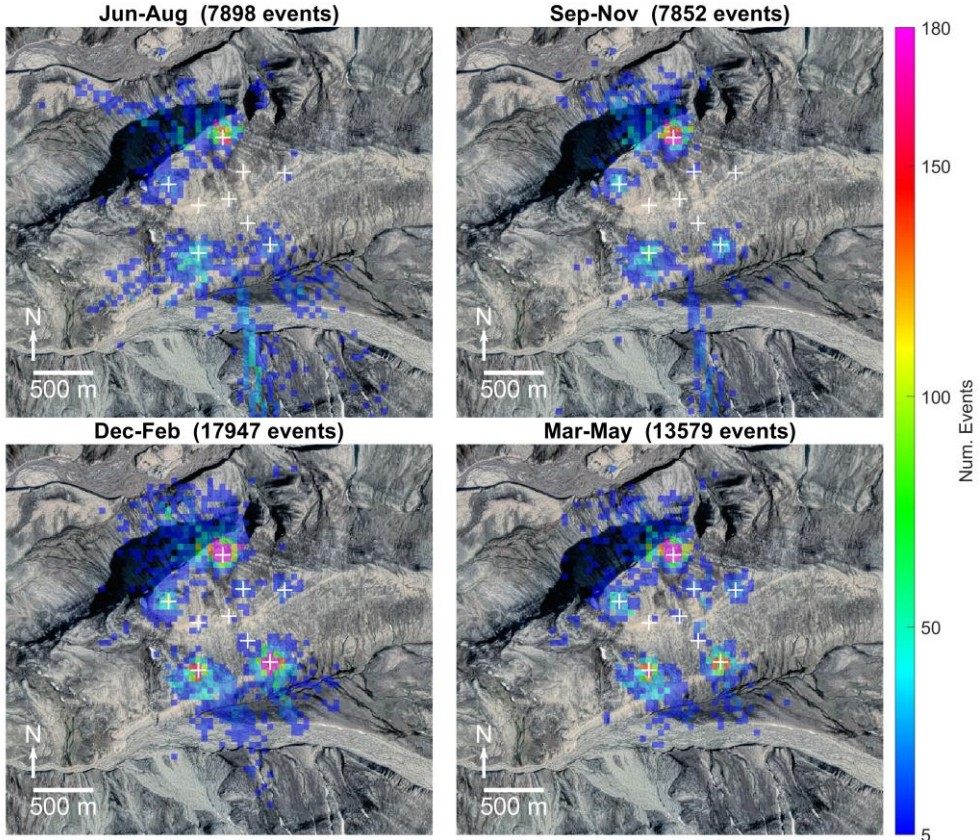

**Figure 7 – Detail view of Janssonhaugen overlaid with spatially binned (50×50 m) distribution of coherent MFP inferred seismic**
**source positions plotted by season for events recorded between August 2004 and July 2021. Positions of SPITS seismometers are indicated by white crosses. Orthophoto © Norwegian Polar Institute (npolar.no).**

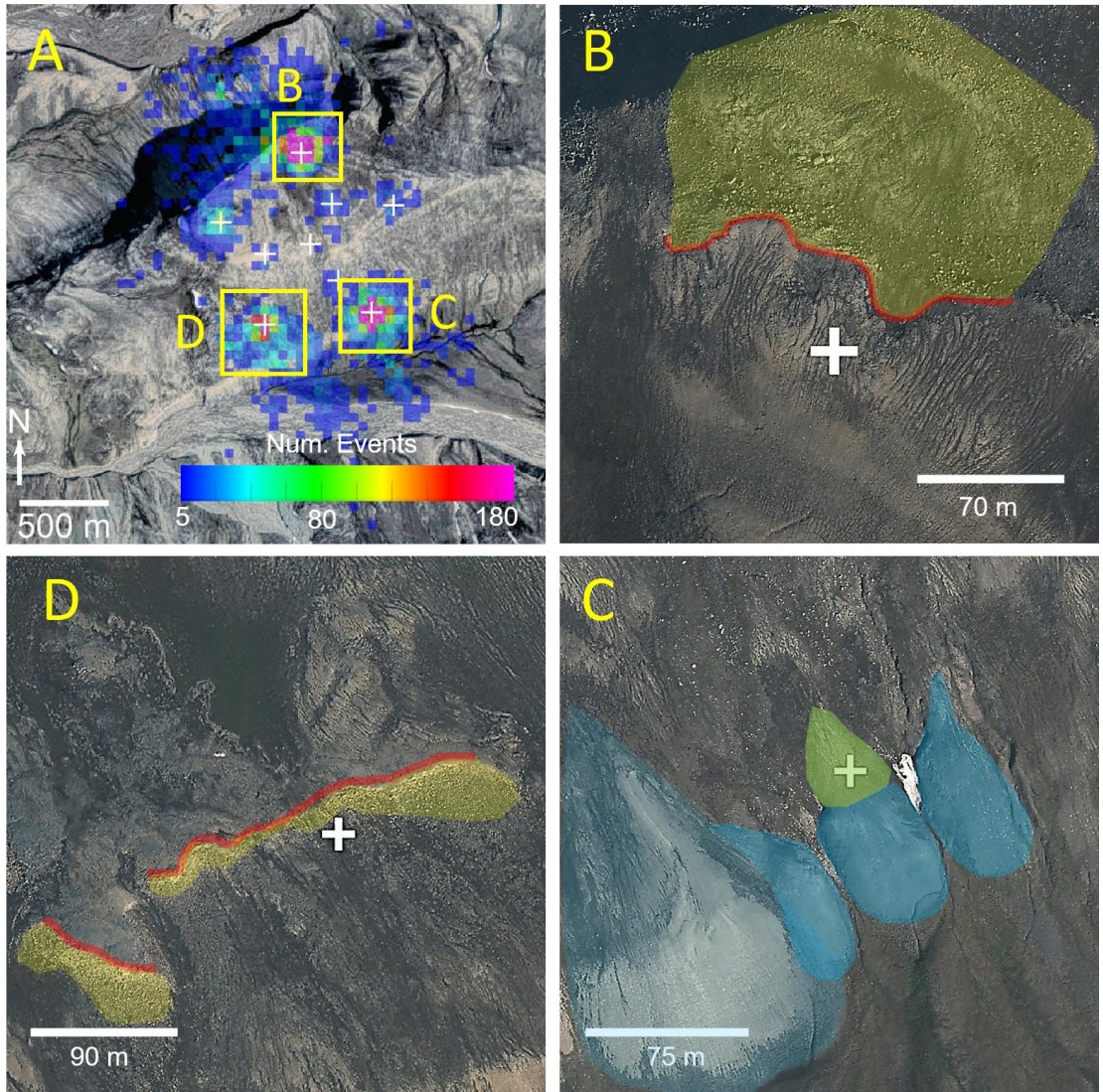

Figure 8 – (a) Dec-Feb events as plotted in Figure 7 with orthophotograph details illustrating the geomorphologic features associated with the most seismically active areas. Erosional scarps (b), (d) are annotated in red and associated boulder fields in yellow, and frozen debris/solifluction lobes (c) are annotated in blue. A faintly visible area of polygonal patterned ground is annotated in green in (c). Orthophoto © Norwegian Polar Institute (npolar.no).

We observe a distinct seasonality for Class I events, with the highest detection rates occurring in the winter months (Dec-Feb) as illustrated in Figure 7. Detection rates are also high during the cold high-Arctic spring (Mar-May) and are lowest during the thaw season. We also observe a cluster of events south of SPITS, with highest activity in the summer months, decreasing in the autumn (Sep-Nov) and absent during the winter and spring. These events may be rockfalls related to fluvial undercutting of steep river cliffs, rockfalls from steep mountain flanks, active-layer detachment slides/debris flows or glacier/rock glacier movements. Using InSAR, Rouyet et al. (2019) measured high summer subsidence rates in the river valley, glacier/rock glacier

and mountain flank areas south of Janssonhaugen corresponding to the inferred source locations of these seismic events. However, since the dynamics of these processes are not represented by our model, we were careful to exclude these events when spatially isolating the cluster interpreted as Class I events. By selecting the subset of events with inferred source positions within ~1500 m of the array centroid, excluding the river valley/rock glacier area south of the array, we isolated a total of 42,432 Class I events recorded between July 2004 and July 2021. Class I events occur most frequently during the cold winter and spring seasons, suggesting a relation to freezing processes.

Locally, the Class I seismicity is dominated by three source areas (see Figure 8) corresponding to areas with erosional scarps and solifluction lobes (Tolgensbakk et al., 2000). The erosional scarps, particularly the one on the northern side of the array (Figure 7), in addition to the steep NW flank of Janssonhaugen, were also active during the summer thaw season. These thaw season events may be rockfalls or other mass movements, possibly initiated by the melting of ice causing loss of strength or joint lubrication (Matsuoka, 2019; Weber et al., 2017). Additional three-dimensional perspective views illustrating the three dominant source areas are provided in Appendix A.

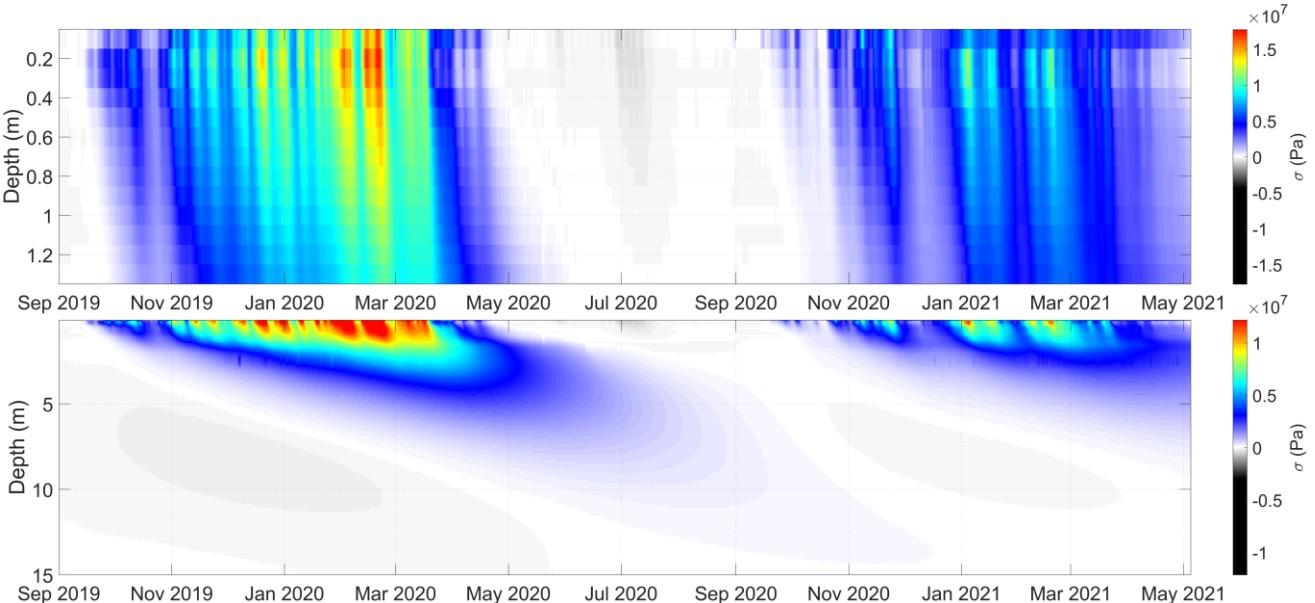

**Figure 9 – Illustration of spatiotemporal thermal stress field modelled according to Eq. (12) and constrained by temperature measurements recorded by sensors installed at Janssonhaugen Vest (0.1 m) and in the P11 borehole (0.2-15 m). The temperature-depth profile used to calculate thermal stress was interpolated to regular 10 cm intervals using a spline interpolant. Sign convention is positive for tensional stress.**

## 4.2 Modelled thermal stress

Figure 9 illustrates the spatiotemporal thermal stress field that was modelled by solving Eq. (12) using the parameters listed in Table 1, the 0.1 m ground temperature timeseries recorded at the Janssonhaugen Vest meteorological station and the 0.2-15 m temperatures recorded in the P11 borehole. The largest thermal stresses occur in the active layer, which is subject to large

amplitude winter cooling cycles. Since the peak annual stresses occur at 0.2 m depth and we have a much longer record from the P11 borehole compared to the Janssonhaugen Vest meteorological station, we focus on this horizon as the dominant thermal contraction cracking depth throughout the rest of the study. That peak thermal stress is modelled at 0.2 m is interesting as it corresponds with the 20-30 cm thick regolith layer at Janssonhaugen (Isaksen et al., 2001), suggesting that cryoturbation within the active layer may have weathered the bedrock over time to produce the surficial layer. Aggradation of weathering material

would be another explanation, but this is less likely since the top of Janssonhaugen is a mountainous plateau where erosion is expected to dominate over deposition. Frost polygons in this region are interpreted to be very old (Sørbel and Tolgensbakk, 2002) so there has likely been sufficient time to reach a steady state condition. If the highest thermal stresses occurred much deeper, we would expect that repeated thermal contraction cracking over thousands of years would have produced a thicker regolith layer than is observed.

**4.3    Thermal stress associated with seismic events**

Figure 10 illustrates the shallowest measured temperature profiles and corresponding thermal stress for the Sep 2019 to Jun 2021 period. The Class II events, corresponding to mining activities at Gruve 7, have source ranges of 6000-7000 m with a consistent azimuth of ~210° and no correspondence with thermal stress. On the other hand, Class I events have variable azimuths, source ranges <1500 m and tend to be associated with peaks in thermal stress.


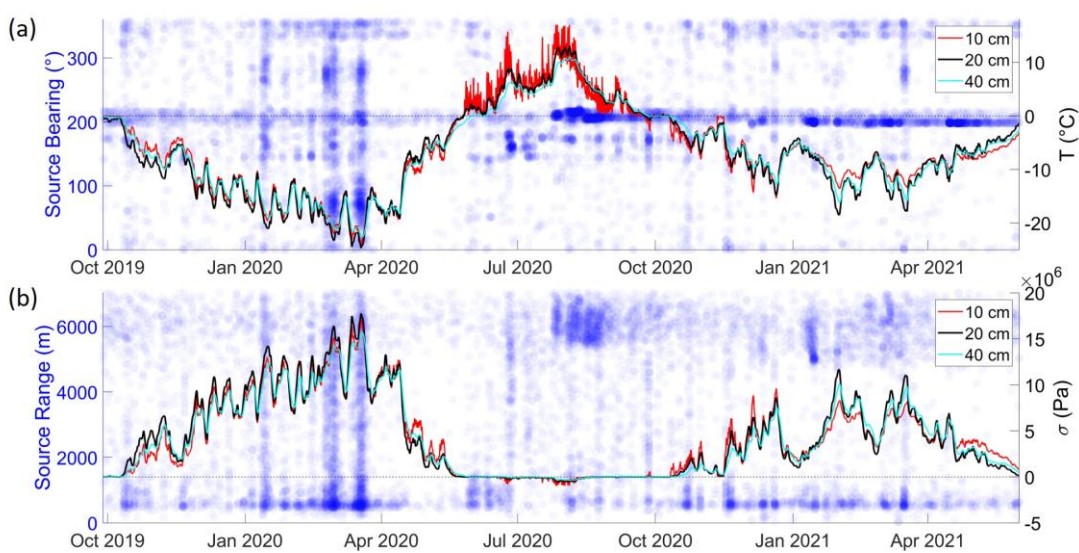

**Figure 10 – (a) Ground temperature profiles (solid lines) and bearing (clockwise from North) to coherent MFP localised source positions from centre of SPITS array (blue circles). (b) modelled ground thermal stress from Eq. (12) (solid lines) and ranges to coherent MFP localised sources (blue circles). Source ranges and azimuths are transparent such that denser colours represent**
**clusters of events.**

Figure 11 shows a detailed comparison of the Class I seismicity and the modelled frequency of tensile fracturing due to thermal stresses exceeding the assumed tensile strength of polycrystalline ice (as described in Section 3.3.1). Given that seismicity is a complex, stochastic process in time and space, our simple thermal stress-based fracture model does a reasonably good job of capturing the time periods and approximate frequency of the Class I events. This leads us to infer that thermal contraction

cracking of ice wedges and crack filling vein-ice, as modelled by Eq. (12) is a significant process contributing to event Class I seismicity. The clusters of events recorded June-August, when thermal stress is low (see Figure 7 and Figure 11), are most likely mass movements associated with steep terrain (e.g., rockfalls, active layer detachment slides, debris flows, etc.), possibly initiated by melting of fracture-filling ice leading to loss of strength or joint lubrication (Matsuoka, 2019; Weber et al., 2017).

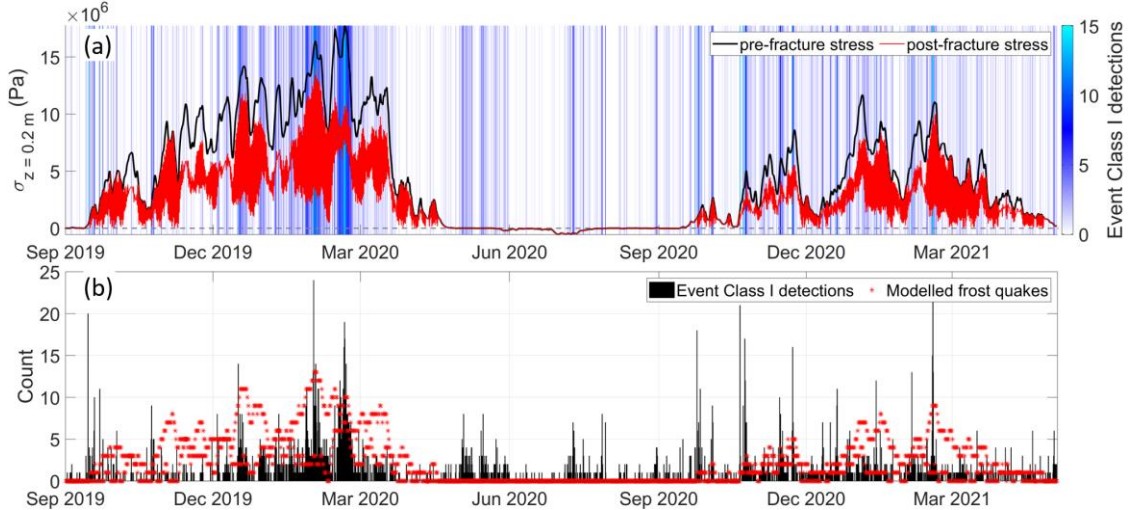

**Figure 11 – (a) Pre-fracture thermal stress (black) given by the solution to Eq. (12) and post-fracture thermal stress accounting for stress release by cracking (red) with blue colour gradient indicating the 6-hourly binned detection rate of Class I events. (b) modelled number of frost quakes (red) according to fracture model (see section 3.3.1) and 6-hourly binned frequency histogram of Class I events (black).**

A similar association between Class I events and peaks in thermal stress persists over the entire study period, from 2004-2021 (see Figure 12). Figure 13 shows that the modelled and observed frost quake seismicity also matches quite well over the study period (the normalised cross correlation is 0.61), though the observed seismicity has a tendency to occur in more defined periods than predicted by the model, resulting in a relatively spiky frequency histogram. Examples of anomalous seismicity not explained by the model are the periods 8-9 Jan 2008, 14-21 Feb 2010, 9-16 Feb 2012, and 5-12 Jan 2016 (see Figure 13).

In order to explain these anomalies, we note that the Feb 2010 episode corresponds temporally to the "C10" major cracking episode identified by Matsuoka et al. (2018) at a field site down-valley in Adventdalen. Matsuoka et al. (2018) observed that this cracking episode was preceded by a highly unusual period of mild weather, accompanied by rain, positive air-temperatures, significant snowmelt and surface water pooling. This surficial water subsequently froze when air temperatures dropped and an extensive series of fresh cracks were observed in the surficial ice by Matsuoka et al., during a field visit on 28 Feb 2010. This

period of anomalous seismicity can therefore be explained as thermal contraction cracking of the surficial ice formed after a period of heavy rain and this explains why they were not accurately predicted by our subsurface thermal stress model (since they occur in response to air temperature rather than ground temperature).

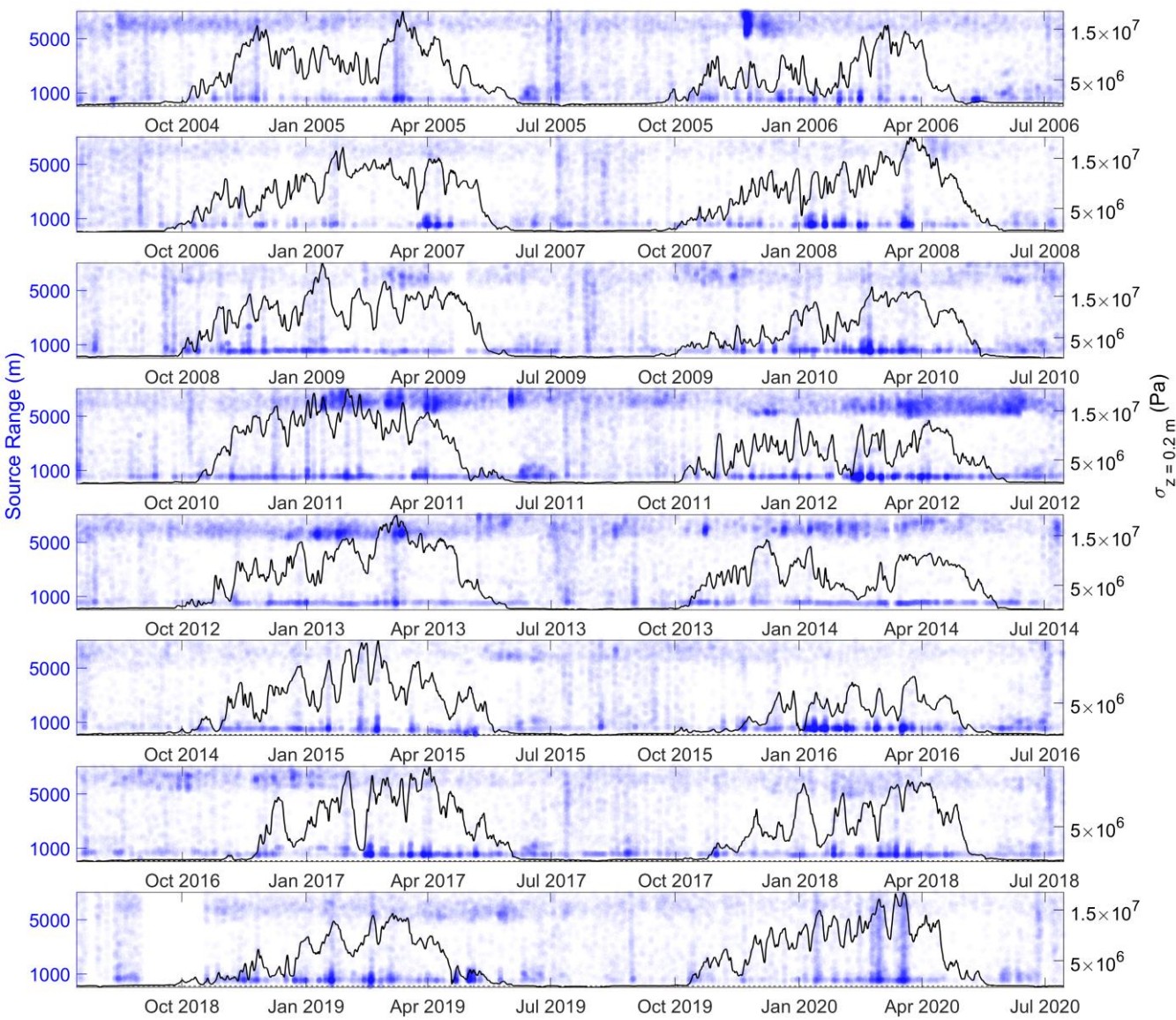

**Figure 12 – Modelled thermal stress from Eq. (12) at 0.2 m depth (black lines) and ranges to coherent MFP localised sources (blue circles). Source ranges and azimuths are plotted as transparent points such that denser colours represent clusters of events. Event Class I & II are associated with small (<1500 m) and large (>5000 m) source ranges, respectively.**

Following this line of reasoning, we were able to connect (see Appendix B) all of the large, anomalous spikes in seismicity that were not predicted by our model with rare, heavy-rainfall events reported on by Dobler et al. (2019). These unusual winter

rainfall events are driven by strong south to south-westerly atmospheric flows with advection of water vapor from warmer areas and are often linked to "atmospheric river" features in the precipitable water anomaly field (Serreze and Stroeve, 2015). That the anomalous spikes in seismicity consistently occur in the wake of significant mild weather and rain events (see Figure B1) further strengthens the interpretation that thermal contraction cracking of newly formed surface ice is a plausible explanation for the anomalous spikes in seismicity we observed at the SPITS array.


We consider the use of real ground temperature measurements a strength of this study, since the subsurface temperature field is a complex product of sensible, latent and convective heat fluxes, depending on matrix and pore-filling material properties as well as surface properties like the thermal conductivity of snow (e.g., Badache et al., 2016; Rankinen et al., 2004). However, it is important to recognize that the P11 borehole and Janssonhaugen Vest ground temperature measurements are point samples

of a temperature field that may vary spatially according to local geomorphology and variation in snow cover. For example, Abolt et al. (2018) have demonstrated that ground under the elevated rims of frost polygons cools significantly faster than the depressed centres due to decreased snow cover. Accounting for variation on the local scale fell outside the scope of the present study, but extending the model framework to allow stochastic temperature variation via stochastic differential equations may give additional insight into the significance of local scale variability.


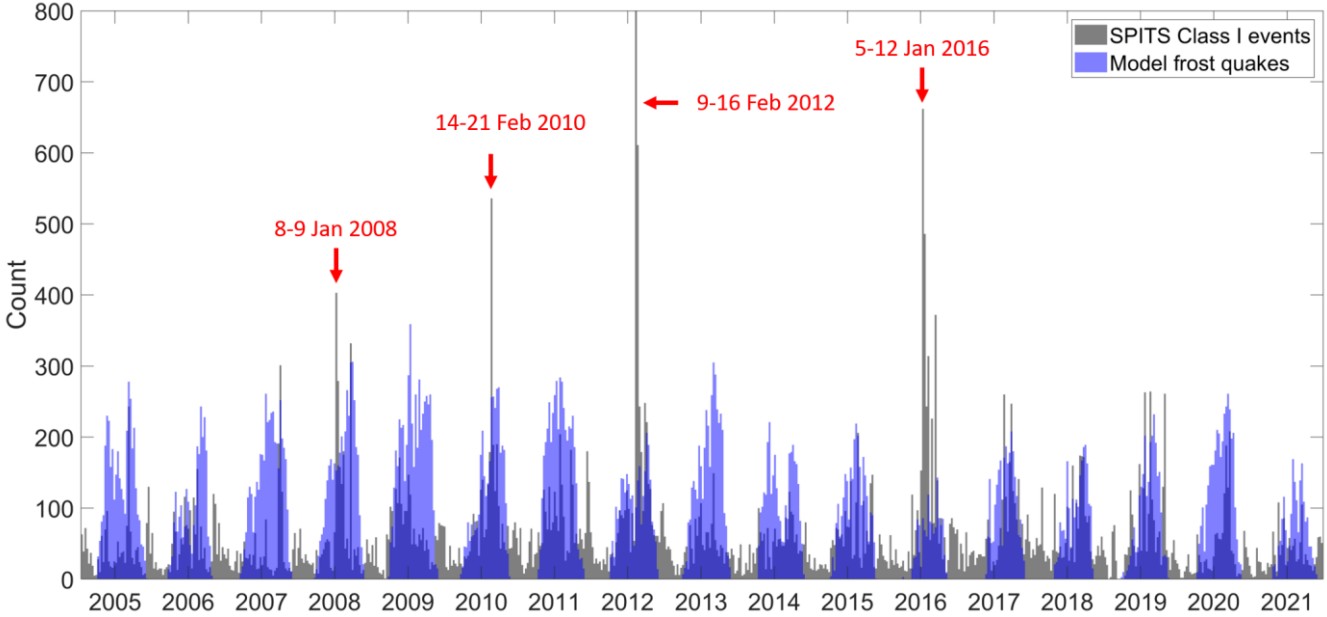

Figure 13 – Histogram binned to 9-day intervals comparing the modelled frost quake frequency (blue) with the recorded frequency of Class I events (black). Anomalous spikes in seismicity following heavy rainfall events (see Figure B1) are annotated in red.

## 4.4 Interpretation of event Class I cryoseismic source

Polygonally patterned ground indicative of cryoturbation of the active layer and/or ice wedges in the underlying permafrost are observed extensively across Janssonhaugen, except where downslope mass movements destroy or interrupt the formation of polygonal networks (Sørbel and Tolgensbakk, 2002). In this study, we observed anomalously high cryoseismicity associated with erosional scarps and a frozen debris/solifluction lobe, which are all associated with downslope mass wasting. As shown in Figure 14, we suggest that these downslope mass movements initiate or precondition transverse cracks or fissures to open

(Darrow et al., 2016; Price, 1974), particularly where there is a transition in slope, i.e., convex terrain. Water from rain or snowmelt infiltrates these fissures and freezes when temperatures drop below freezing (e.g., Darrow et al., 2016). Rapidly falling ground temperatures during winter then cause the surrounding ground to thermally contract. This causes the vein-ice filling the frozen fissures to crack under tension, since the tensile strength of ice is lower than the surrounding ground and therefore constitutes a plane of weakness (e.g., Plug and Werner, 2002). Cracking relieves the accumulated thermal stress.


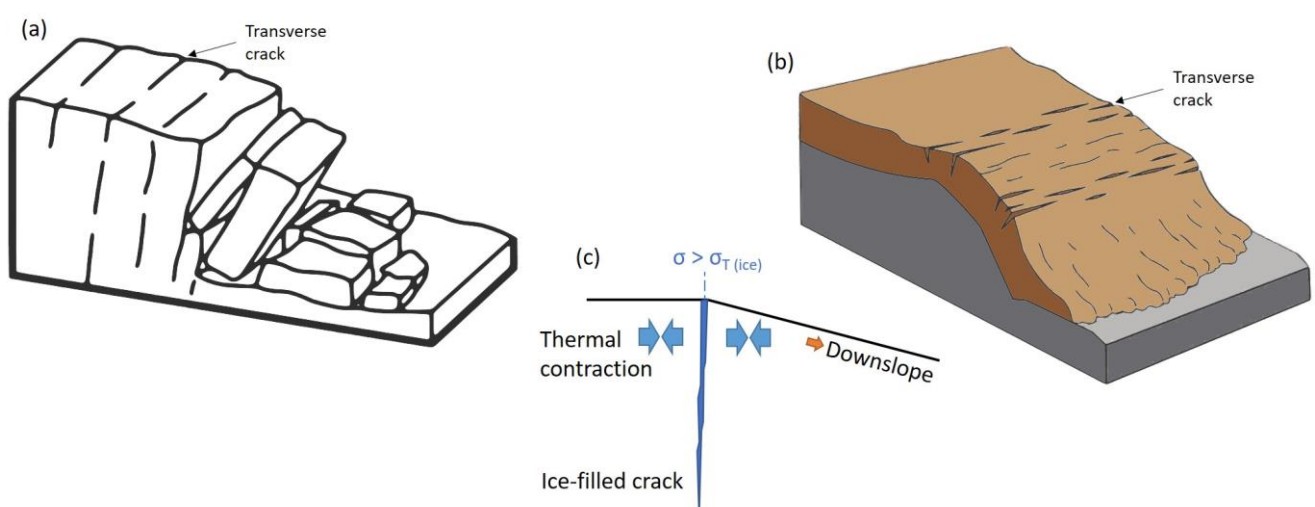

**Figure 14 – Mass movements such as, (a) erosional scarps and (b) frozen debris/solifluction flows initiate transverse cracks/fissures which become infiltrated by water that freezes to ice. (c) Thermal contraction of the surrounding ground and downslope gravitational stress causes the ice to crack under tension when the stress, σ, exceeds the tensile strength of ice, σ$_{T(ice)}$.**


This mechanism is analogous to the Lachenbruch (1962) model of thermal contraction cracking of ice wedges in permafrost (see Figure 1). However, the cracking may occur more frequently or under milder surface cooling because 1) the frozen fissures may be pre-stressed by downslope gravitational forces making them more prone to failure and 2) the fissures extend to the ground surface where thermal stresses are largest. For the case of ice wedges, the ice wedge is located below the permafrost

table (Figure 1c). There may be vein ice extending through the active layer only if the previous seasons thermal contraction crack has remained open through the summer thaw season. If the crack has closed so that vein ice is not formed during the

early freezing season, initiation of thermal contraction cracking of the ice wedge would require accumulation of thermal contraction stress at the level of the permafrost table, requiring a longer or more extreme surface cooling episode. Tensile cracks could also form in the active layer where ice veins/ice wedges are absent (as in Figure 1a), but this would require thermal contraction stresses exceeding the tensile strength of frozen ground, which is greater than that of polycrystalline ice (e.g., Plug and Werner, 2002) and may therefore occur less frequently.

## 5    Conclusion

We studied the spatial and temporal patterns of the class of seismicity associated with short duration ground shaking at the SPITS array in Adventdalen, Svalbard, based on a catalogue of >100 000 events recorded between 2004 and 2021. To the best of our knowledge, this is a uniquely large and long spanning event catalogue amongst studies with a focus on cryoseisms. We find that these short duration seismic events, with ground motion lasting just a few seconds, can be grouped into two main subclasses. One class is clearly associated with mining activities at the underground coal mine, Gruve 7 and is mostly useful as an indicator of the performance of the coherent-MFP source localisation algorithm. The other subclass of short duration seismic events is interpreted to be dominated by frost quakes produced by thermal contraction cracking of ice wedges and crack filling vein-ice. These events appear to be dominated by surface wave energy and source positions that are proximal to the SPITS array, particularly three areas that are associated with dynamic geomorphological features; erosional scarps and solifluction lobes. Temporally these events are associated with peaks in ground thermal stress as modelled by a simple dynamical thermo-viscoelastic model constrained by borehole measurements of ground temperature. The long-term continuous observational record, containing tens of thousands of inferred cryoseismic events, in close proximity to high-quality borehole temperature observations, provides a unique insight into the spatiotemporal patterns of cryoseismicity. Further experiments using controlled test sources in known locations could be used to calibrate MFP model assumptions that could subsequently benefit future MFP analyses of the complete, long-term seismic record from SPITS.

## Appendix A.        3D perspective models of study area

Three-dimensional perspective views of the study area were rendered by draping a composite orthophoto over a digital elevation model. Figure A1 shows the erosional scarp and frozen debris/solifluction lobe on the southern flank of Janssonhaugen that were associated with spatial peaks in Class I seismicity. Figure A2 illustrates the second erosional scarp on the north-eastern flank of Janssonhaugen that was also associated with a peak in Class I seismicity.

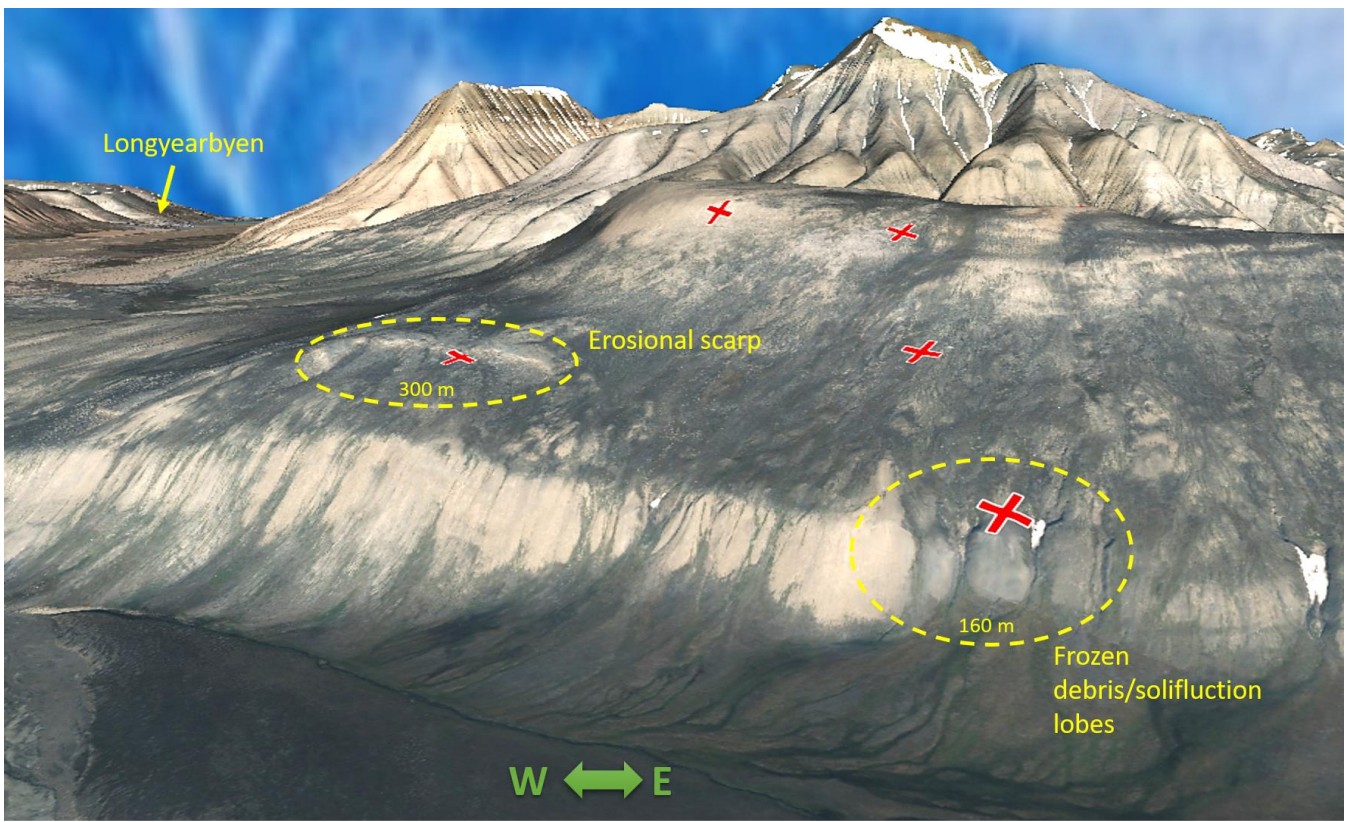

**Figure A1 - 3D perspective view of the southern flank of Janssonhaugen highlighting two of the three areas associated with anomalous seismicity. The 3D model was constructed by draping an orthophoto on top of a 1.5x vertically exaggerated DEM. Red crosses mark locations of SPITS seismometer stations. Orthophoto/DEM © Norwegian Polar Institute (npolar.no).**


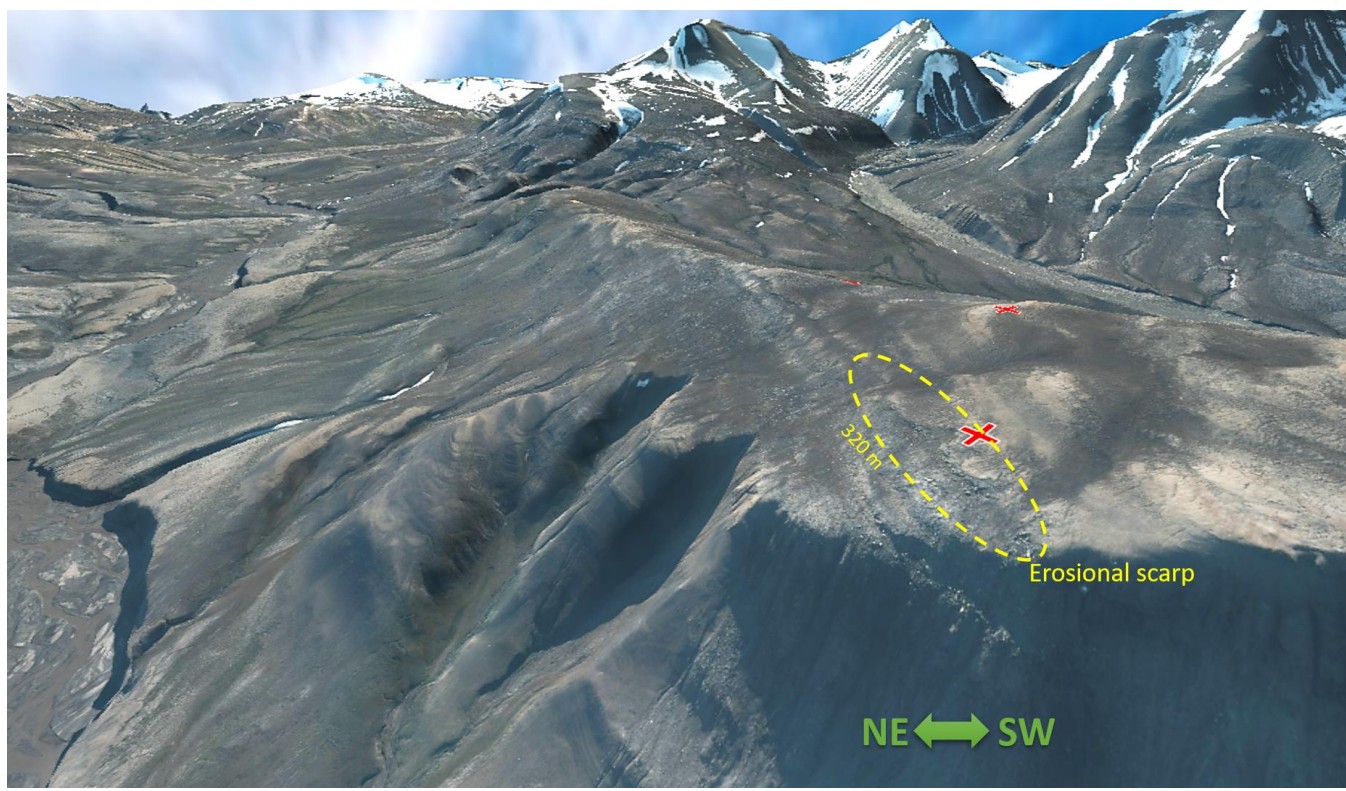

**Figure A2 - 3D perspective view of the north-eastern flank of Janssonhaugen highlighting the third area associated with anomalous seismicity. The 3D model was constructed by draping an orthophoto on top of a 1.5x vertically exaggerated DEM. Red crosses mark locations of SPITS seismometer stations. Orthophoto/DEM © Norwegian Polar Institute (npolar.no).**

**Appendix B.     Meteorological conditions associated with anomalous seismicity**

Figure B1 illustrates the meteorological conditions preceding anomalous spikes in seismicity that were not captured by the subsurface thermal stress and simple fracture model presented in this study. Meteorological conditions are represented by temperature and precipitation observations from Svalbard airport (~21 km from Janssonhaugen), since these variables were not recorded at Janssonhaugen over the relevant time period. The meteorological observations show that the anomalous periods of seismicity were preceded by >0 °C temperatures and rain in all instances. We expect that these rainfall events led to surface water pooling that subsequently froze to ice when temperatures again fell below 0 °C. The periods of rapid cooling occurring synchronously with the anomalous seismicity would have then promoted intense thermal contraction cracking of the exposed surficial ice.

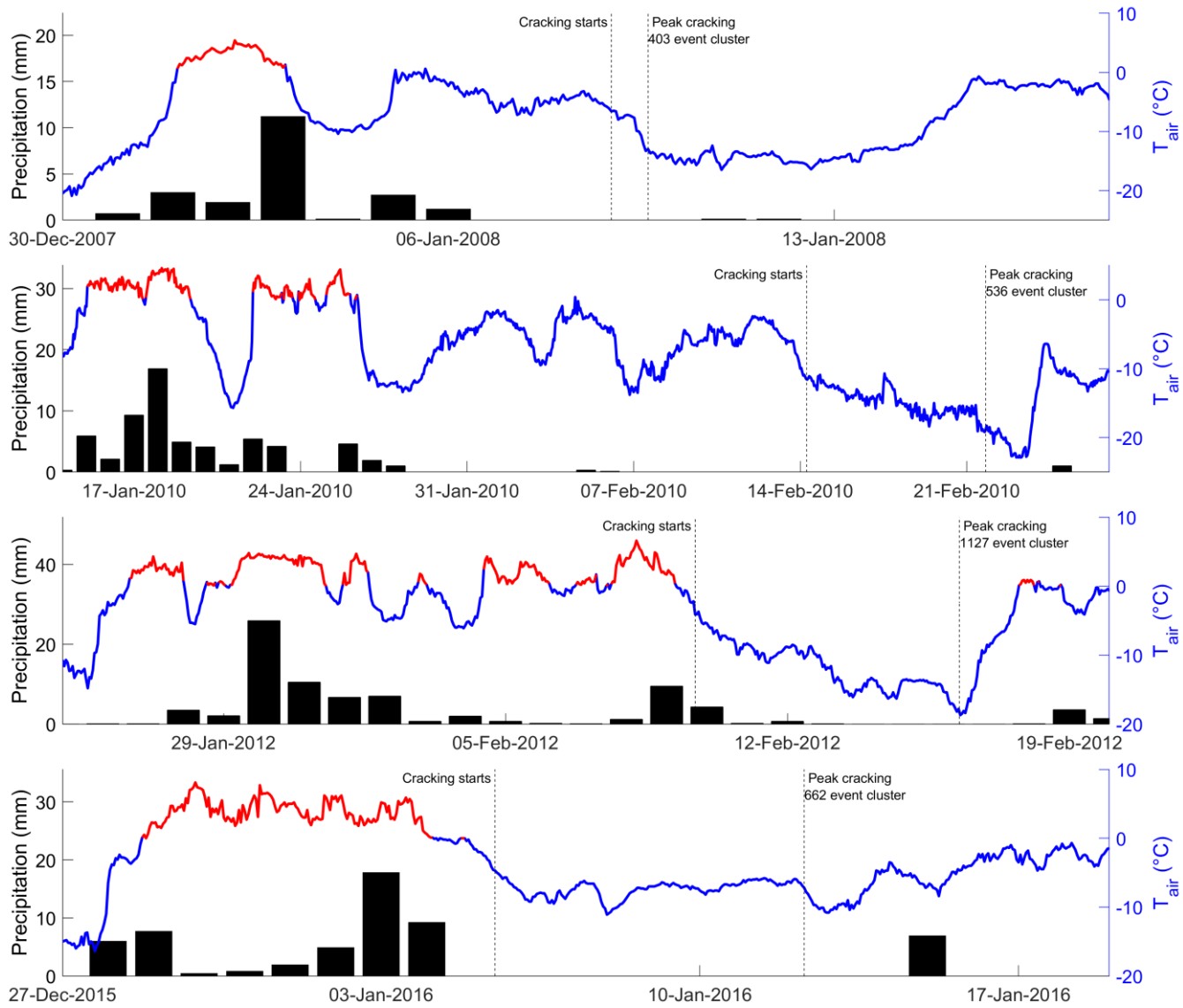

**Figure B1 - Svalbard airport air temperature (line) and daily precipitation (bar) records (met.no) illustrate that anomalous spikes in cracking related seismicity were preceded by unseasonably mild temperatures >0 °C (red line) and rain, presumably leading to snow melt, water pooling and surface ice formation as observed in the field at Adventdalen by Matsuoka et al. (2018) in Feb 2010.**

**Appendix C.    Additional details on signal characteristics of detected seismic events**

Two classes of short duration seismic events were identified. Event Class I was interpreted to be dominated by cryoseisms while event Class II was associated with mining activities at the operational coal mine, Gruve 7. Figure C1 shows that the dominant signal frequency is (on average) lower for Class II than for the SPITS proximal Class I events. We also see clustering at distinct dominant frequencies for event Class II, probably related to differences in the specific mining activities that trigger

these events. Event Class I spans a wide range of dominant frequencies, though the most common is ~17 Hz (see Figure C1-

a). Bandwidth was estimated as the maximum signal bandwidth across all array stations (Figure C1-b-c). The bandwidth

containing 99% of the signal energy is quite high, since it is typically around 30 Hz and the waveforms are pre-filtered with a

5-35 Hz passband. On the other hand, the half-power (3 dB) bandwidth is quite low, rarely exceeding 5 Hz. This matches our

observation that even though Class I events appear to be dominated by surface wave energy (which is typically dispersive),

we don't observe the distinctive highly dispersive waveforms that one would expect for a wideband dispersive signal (see

Figure 5).

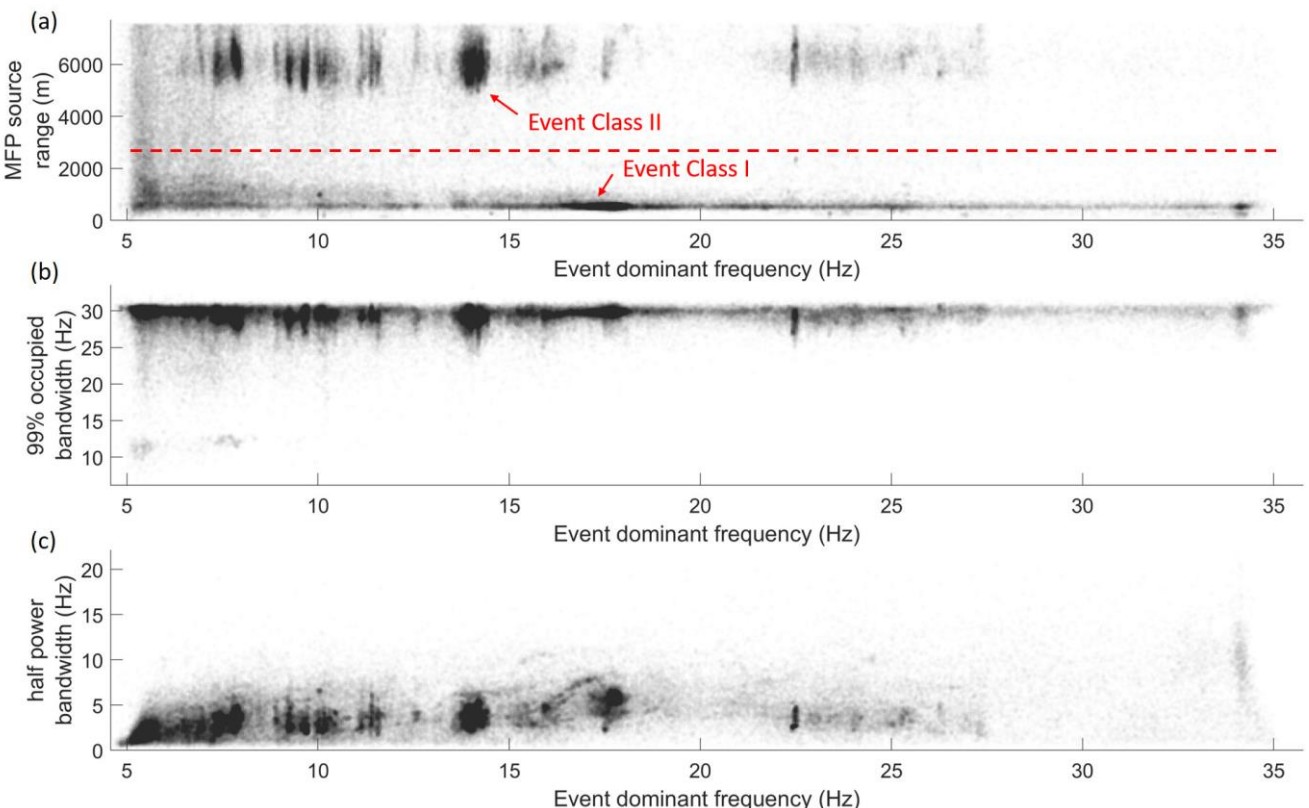

**Figure C1 - Illustration of relationship between dominant frequency estimated by Thomson's multitaper spectral amplitude peak and (a) MFP estimated range to source, (b) bandwidth containing 99% of signal power estimated from periodogram power spectral**
**density and (c) half-power bandwidth determined as the frequency range where power spectral density is within 3 dB of the maximum. The two event classes are most readily identified by the range to source, as annotated by the red dashed line.**

Figure C2 shows illustrative examples of three component seismograms for the two identified event classes. There is some

indication of phase rotation between the vertical and radial components of ground motion for the highest amplitude arrival of

event Class I (frost quake), which would indicate a Rayleigh wave. The Class II event (mining activity at Gruve 7 coal mine)

that is interpreted to be dominated by P wave energy has vertical and radial components that are more similar to one another,

as expected for body waves. In addition, the largest amplitudes for the Gruve 7 event correspond to the first arrival (consistent with interpretation as P-wave dominated), whereas the largest amplitudes for the frost quake arrive later (consistent with interpretation as Rayleigh wave dominated).

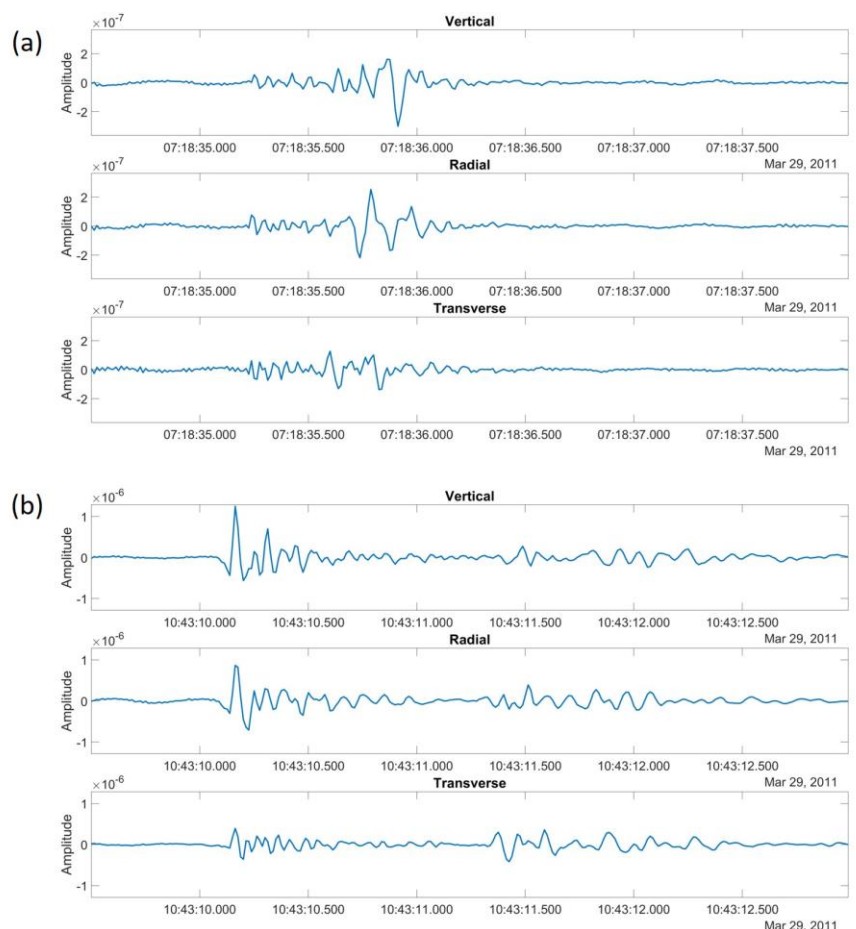

**Figure C2 - Three-component seismograms from SPITS station SPA0 for (a) an example of event Class I (frost quake) and (b) an example of event Class II (Gruve 7 mining activity). The signals are bandpass filtered to 2.5-35 Hz and the horizontal components of ground motion (measured in NS and EW orientations) were rotated to radial and transverse to the source bearings estimated by MFP (see Figure 5e and Figure 5f).**

## Appendix D.     Frequency bandwidth and MFP

The MFP results in this study were found to be insensitive to the frequency band analysed, particularly for the coherent-MFP formulation, Eq. (7). Figure D1 shows that as the passband is narrowed, resolution decreases somewhat for coherent-MFP but the estimated source position remains more or less constant. The incoherent-MFP (classical formulation, Eq. (4)) performs poorer throughout and is progressively more biased towards station number 8 as the passband is narrowed. We can reject this

result because Figure D1-a clearly shows that the signal arrives at station 9 around the same time or slightly before station 8,

so the incoherent-MFP maximum in the vicinity of station 8 (Figure D1-i) must be spurious and is likely a result of interfering

sidelobes whose position varies with frequency.

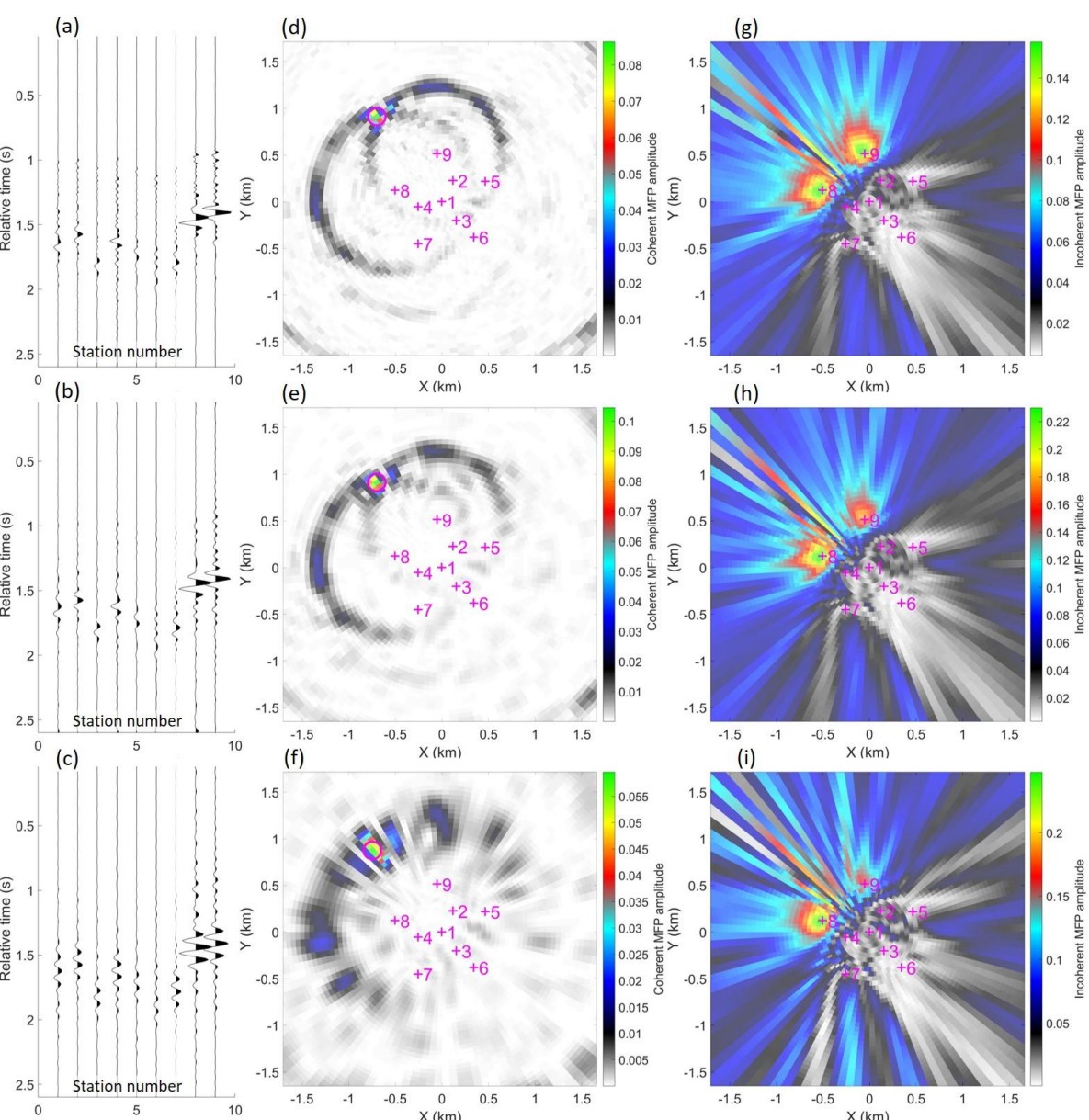

**Figure D1 - Example of MFP source location for an event with dominant frequency of 10.2 Hz and half-power bandwidth of 6.7 Hz**
**(same event as shown in Figure 5b) filtered to passbands of (a) 5-35 Hz, (b) 5-15 Hz and (c) 7.5-12.5 Hz prior to MFP. (d), (e) & (f)**
**show corresponding coherent MFP ambiguity surfaces while (g), (h) and (i) show incoherent MFP ambiguity surfaces.**

## 6    Code/Data availability

All data used in this study is publicly available, seismic waveform data may be downloaded via http://eida.geo.uib.no. The Janssonhaugen P11 temperature monitoring borehole is part of the GTN-P database (Global Terrestrial Network for Permafrost) and data is available upon request to the custodian. Meteorological data (including ground temperature) from the Janssonhaugen Vest weather station is available via the Norwegian Centre for Climate Services https://seklima.met.no/. The code used to produce this research can be shared upon request to the authors.

## 7    Author contribution

The study was conceptualised by AK and RR. The model, theory and methodology were developed by RR and AH. RR carried out the data collation and processing and was responsible for analysing and visualising the data. RR drafted the initial manuscript with contributions from all authors, with AH contributing significantly to the development and drafting of the theory sections. AH and AK also provided project supervision.

## 8    Competing interests

The authors of this article declare that they have no conflict of interest.

## 9    Acknowledgments

This research is funded by the University of Tromsø - The Arctic University of Norway, by the ARCEx partners and by the Research Council of Norway through grant number 228107. The publication charges for this article have been funded by a grant from the publication fund of UiT The Arctic University of Norway.

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
