# Peer review of "Long-term analysis of cryoseismic events and associated ground thermal stress in Adventdalen, Svalbard"

_The Cryosphere, 2021_

## Referee Comment (RC2)

[referee-annotated manuscript omitted]

---

## Referee Comment (RC3)

**Subject:** Review to: Long Term Analysis of Cryoseismic Events and Associated Ground

**Date:** December 17, 2021
* * *
**1 General Comments**

This article aims to study the hypothesis that short-duration seismic events at SPITS can be due to the frost quakes induced by thermal contraction cracking. The main interest of the study lies in the identification of seismic events that correspond to frost quake initiated by thermal stress. Two event classes were defined by authors and the second class of seismic event was concluded due to the underground mining operations. The first event was found to be active during winter (Dec-Feb) and spring (Mar-May) and inactive in the summer time.

The article is well written and the research is interesting. A contribution from this paper is that the authors used the ground temperature measured by a series of thermistors in the numerical analysis. However, the proof of the hypothesis (especially for the frost action induced seismic events) is still not convincing. In my opinion, the paper needs to be revised to convincingly explain how frost action can initiate the dynamic response. The following comments are also needed to be addressed.

First, the authors should include a fuller explanation of the physical mechanism of frost quakes and how the frost action could induce dynamic responses.

Second, the classification of two classes of seismic events must be addressed more quantitatively, in order to classify and detect frost action related seismic events.

Third, the mismatch between the wavelength of seismic events (could be more than 115 m) and the studying depth (top 15) needs to be better explained or addressed.

**2 Specific Comments**

Line 32: I would add a few sentences to explain why ground cooling induces cracking. Is it because of the volumetric expansion during the phase change from water to ice? Or it can be due to the formation of ice lenses which is associated with the water migration during the freezing process? Can also thawing contribute to cracking?

Line 34: Frost heave is an upward swelling of soil due to an increasing presence of ice as it grows towards the surface (continuously delivers water to the freezing front via capillary action). I am having difficulty understanding that frost heave is 'rapid' and also 'elastic' deformation. The frost heave requires delivering unfrozen water contentiously to the freezing front via capillary force, which is likely a slow process (Darcy's Law). It is not necessarily elastic either given its large deformation.

Line 49: What is the mechanism for the segregation ice growth in bedrock? If it is the same as the frost heave, where does the capillary force come from?

Line 119: It would be useful to also provide the temperature distribution in the permafrost at the location of P11 (even in the supplementary information).

Line 131: Authors need to provide more explanation to illustrate why cryoseismic events have a shorter duration than other seismic events.

Line 136: It would be useful providing any physical interpretations of the ratio of short-time-averaged amplitude and long-time-averaged amplitude (explain why this ratio can be used to detect cryoseismic events). I would also add a few sentences to explain why choose Hilbert transform for the short-time-averaged amplitude and moving average for long-time-averaged amplitude.

Line 141: Authors stated that 'By visual inspection of test periods, we found that this emphasizes very local events with large amplitude variation across the array, while still ensuring that there is at least some coherency across the array'. It would be better to explicitly indicate what 'this' represents. More importantly, there should be a figure to show what authors captured by their visual inspection and prove how it emphasizes local seismic events.

Line 151: Authors concluded that the ratio of short-time-averaged and long-time-averaged peaks must have amplitude larger or equal to 10 and occur at least 5 seconds apart from one another. Authors need to explain how they drew this conclusion, or clearly indicate this is an assumption if that is the case.

Line 209: It is common to add the unit for every applicable parameter defined in the manuscript.

Line 253: Authors stated that 'A significant novelty of this study is that the ground temperature profile at Janssonhaugen has been logged by a series of thermistors installed in the 15 m deep P11 borehole at 6 hr intervals since April 1999'. How did the authors use these measurements in the analysis in the entire study domain (I suspect interpolation is required)?

Line 254: Authors need to elaborate: 'We assume the stress at a given depth is decoupled from the stress at adjacent depths'. How exactly did the authors decouple the stress components in space since they are dependent on each other (if assuming the soil is continuum media).

Equation 10: Can this model be used to study the thawing of frozen soils?

Table 1: How the tensile strength is calculated? Is it always 1 MPa? Should it also be temperature-dependent?

Line 288: Authors stated that 'Event class I is characterised by significant amplitude variation and arrival time differences across the array seismometers'. A quantification method (e.g., L2 distance) is needed to describe the significant amplitude variation as well as similar amplitudes (line 291). Also, authors predicted the source of seismic events is around 1500 m. What is the uncertainty of this prediction?

Line 299: Authors stated 'The mean MFP inferred propagation velocities for Class I events was 1150 m/s with a standard deviation of 1100 m/s, indicating that they are dominated by surface waves. The large standard deviation may indicate the surface waves are dispersive with different frequencies propagating at different phase velocities.' In line 132, the authors also mentioned the signal is filtered to a range of 2.5-20 Hz. This gives us a wavelength around 115 m (1150 m/s divided by 10 Hz, average frequency) and the investigation depth in this paper is only about the first 15 m. The authors should explain this mismatch.

Line 303: It is difficult to determine the dominant wave type based on merely the estimated propagation velocity. Could the 1150 m/s also correspond to body wave?

Figure 3: It is difficult to understand why longer seismic events have high amplitude. Authors might want to elaborate on the relation between the duration and the amplitude of displacement (or velocity and acceleration).

---

## Author Comment (AC1)

- Detected events and their location:

1. I think the nature of the events should be better introduced. In particular, the frequency context is not discussed, nor is clear, which frequency range is actually analyzed. The text mentions, that the STA/LTA detector is applied to 2.5-20 Hz bandpass filtered data, while the location procedure is mentioned to happen in the 5-35 Hz band. I suggest to add details on the event's frequency content and on the frequencies used.

This is a good point. Recalling that event Class II is associated with relatively large MFP inferred source ranges, we see that the dominant frequency is (on average) lower than for the SPITS proximal Class I events (see Figure I). We also see clusters at distinct frequencies for event Class II, probably related to differences in the specific mining activities that trigger these events. Event Class I (interpreted as cryoseisms) span a reasonably wide range of dominant frequencies, though the most common seems to be ~17 Hz (see Figure I). We estimate the bandwidth as the maximum signal bandwidth across all array stations. The bandwidth containing 99% of the signal energy is quite high, since it is typically around 30 Hz and the waveforms are pre-filtered with a 5-35 Hz passband. On the other hand, the half-power (3 dB) bandwidth is quite low, rarely exceeding 5 Hz. This matches our observation that even though Class I events appear to be dominated by surface wave energy (which is typically dispersive), we don't observe the distinctive highly dispersive waveforms that one would expect for a wideband dispersive signal (see Figure 4 in the manuscript).

Given that the Nyquist frequency of these data is only 40 Hz, it is difficult to conclude whether the limited bandwidth is due to a physical effect or insufficient temporal sampling. Romeyn et al. (2021) showed that successive modes in a multimodal surface wavefield produced by frost quakes become dominant over frequency bands of ~8-10 Hz before the next mode takes over and becomes dominant. The ~5 Hz half-power bandwidth we observe for the events recorded by SPITS might indicate that a single mode is preferentially excited for a given frost quake, with the mode dominating the 15-20 Hz band being most commonly excited. While it is difficult to address this topic adequately in this manuscript (due to the low Nyquist frequency), it could be an interesting subject for further research. That certain frost quakes preferentially excite different surface wave modes would be an exciting result if it could be convincingly demonstrated. It would be even more interesting if one could develop a model to explain the physical mechanism by which such preferential excitation might occur (fracture geometry, depth, etc. could play a role).

*Romeyn, R., Hanssen, A., Ruud, B. O., Stemland, H. M., and Johansen, T. A.: Passive seismic recording of cryoseisms in Adventdalen, Svalbard, The Cryosphere, 15, 283–302, https://doi.org/10.5194/tc-15-283-2021, 2021.*

[Figure]

*Figure I – Illustration of relationship between dominant frequency estimated by Thomson's multitaper spectral amplitude peak and (a) MFP estimated range to source, (b) bandwidth containing 99% of signal power estimated from periodogram power spectral density and (c) half-power bandwidth determined as the frequency range where power spectral density is within 3 dB of the maximum. The two event classes are most readily identified by the range to source, as annotated by the red dashed line.*

In regards to the different pass-bands used in different aspects of the study. High frequency noise is important to eliminate during initial data screening since it can lead to spurious spikes in STA/LTA. When using MFP to estimate the source location of an event, spatially incoherent high-frequency random noise is not problematic and does not contribute significantly to MFP amplitude. However, If the signal contains spatially coherent high-frequency energy, this will contribute to the MFP amplitude. We found that low-frequency background noise had a higher chance of being spatially coherent. As a result, it was beneficial to eliminate the signal/noise shared bandwidth on the low frequency side, while retaining frequencies containing both signal and noise on the high frequency side for MFP. Bandpass filtering is inherently a trade-off between signal retention and noise elimination and we have attempted to optimize the pass-bands as much as possible according to the peculiarities of the STA/LTA signal detection and MFP source estimation methodologies.

This discussion is too lengthy to incorporate in the main body of the manuscript in its entirety, but we can add some of the key details from this response to sections 3.1 and 4.1 and include the figure as an appendix to the revised manuscript.

2. The authors use a cross-frequency formulation of matched-field processing to locate the seismic events. This approach favors the spatial coherence of the wavefield across frequency. However, event class I in the manuscript is interpreted to be dominated by surface wave energy, in which case dispersion should work against this spatial coherence. In my opinion, it would be interesting to compare the results of this approach with the classical formulation and a more narrow frequency band, e.g. centered around the dominant frequency of the events. In summary, I think that the robustness of the location results should be assessed.

This is also a good point. Surface wave dispersion is unlikely to play a major role for the events we consider since the amplitude spectrum is dominated by a relatively narrow band of frequencies for a given event (see response to point 1). As shown in Romeyn et al. (2021), the dispersion spectra of the multimodal surface wavefield are reasonably flat for a given mode and jump abruptly in phase velocity between modes. That the individual frost quakes recorded at SPITS are dominated by specific individual wave modes may be an explanation for the absence of highly dispersive waveforms. The MFP results are quite insensitive to the specified frequency band, particularly for the coherent-MFP formulation (see Figure II). Figure II shows that as the passband is narrowed, resolution decreases somewhat for coherent-MFP but the estimated source position remains more or less constant. The incoherent-MFP (classical formulation) performs poorer throughout and is progressively more biased towards station number 8 as the passband is narrowed. We can reject this result because Figure II-a clearly shows that the signal arrives at station 9 around the same time or slightly before station 8, so the incoherent-MFP maximum in the vicinity of station 8 (Figure II-i) must be spurious (and is likely due to interfering sidelobes).

While they appear to be robust, the accuracy of the MFP estimated source locations is worthy of further discussion, particularly in response to the reviewer's comment:

> "Line 330, Fig. 6 and especially Fig. 7: Interestingly, the three main source clusters of class I events are centered exactly around three of the array stations. This looks a bit suspicious to me, could this be an artifact in the MFP results, can you comment on this?"

In general, we would consider the MFP results as estimates that are consistent with the evidence available. The approximate vicinity of these clusters must be correct, because we have manually checked many individual events and the estimated source positions are consistent with relative arrival times and amplitudes (i.e., first arrival with strongest amplitude at the closest station, latest arrival with weakest amplitude at the furthest station etc.). However, in detail some degree of clustering/local attractor type behavior is almost guaranteed to occur due to data/model phase velocity mismatches and amplitude attenuation that doesn't exactly match the model. The error in the estimated source position will then be proportional to the amplitude spectrum weighted model/data phase velocity mismatch integrated over the signal bandwidth, convolved with the deviation from a purely geometrical spreading model of amplitude attenuation. It is very difficult to quantify this deviation *a priori* so the best way to assess MFP accuracy is by locating known test sources. As a side note, we found no location bias when locating synthetic test sources where phase velocity and amplitude variation perfectly matched those assumed by the model.

We can add a sentence to the conclusion of the revised manuscript that it would be worth carrying out additional field experiments in the future, using a set of controlled position sources, e.g. sledgehammer on steel plate, to constrain and calibrate the MFP source estimation accuracy across the study domain. While not ideal, the fact that the seismic events associated with mining activities at Gruve 7 show reasonable spatial correspondence with the location of the underground mine workings is encouraging. It's also interesting that there are only three stations that stand out as being associated with anomalously high seismicity (see Figures 6 and 7 in the manuscript). These three clusters hold up well under visual QC of the waveforms in terms of relative arrival time and amplitudes. We can also exclude that the events cluster towards all of the outermost stations in the array because there are very few events associated with the easternmost station (station 5 in Figure 4/Figure II).

[Figure]

*Figure II – Example of MFP source location for an event with dominant frequency of 10.2 Hz and half-power bandwidth of 6.7 Hz filtered to passbands of (a) 5-35 Hz, (b) 5-15 Hz and (c) 7.5-12.5 Hz prior to MFP. (d), (e) & (f) show corresponding coherent MFP ambiguity surfaces while (g), (h) and (i) show incoherent MFP ambiguity surfaces.*

3. The aperture of the array does not seem to be ideal for the analyzed events. Given a minimum interstation distance of roughly 250 m and an aperture of 1 km , as stated in the text, the resolvable wavelength range according to Tokimatsu 1997 (see also Wathelet et al., 2008) is roughly 500-3000 m. Given the frequency range of 2.5-20 Hz (?) and the determined velocities of 1150 m/s (class I) results in considerably smaller wavelengths (while class II events seem well suited for the array aperture). I am not saying this will not work, but there should be some discussion again on the robustness of the results.

Reference: Wathelet, M., Jongmans, D., Ohrnberger, M., & Bonnefoy-Claudet, S. (2008). Array performances for ambient vibrations on a shallow structure and consequences over Vs inversion. Journal of Seismology, 12(1), 1-19.

It is absolutely correct that the station spacing of the SPITS array is not ideally suited to sampling the wavelengths of interest in this study. We have found it very challenging to extract high-quality dispersion curves from these data using the normal phase-shift methods, in line with the Wathelet et al. (2008) study and the reviewer's calculations. This is also not a surprising result because the primary purpose of the SPITS array is to record regional or teleseismic earthquake signals that are dominated by much longer wavelengths. However, a useful property of the MFP method is that the recorded wavefield is "compared" to a model wavefield and the source position is taken as the location giving the strongest coherence between recorded and modelled wavefields. In this way MFP is less susceptible to phase wrapping issues that limit the use of small wavelengths in normal phase-shift dispersion spectrum estimation.

The array layout certainly affects the precision with which source locations can be estimated. However, as mentioned in the previous response, this is as much a function of the data-model misfit as it is a function of bandwidth, the number of array elements and their spacing. We think Figure 4 in the manuscript provides a good overview of the lobe pattern that is observed in practice for seismic sources at different ranges and azimuths (these plots are often referred to as MFP localization ambiguity surfaces). As discussed in the following response (point 4), the carefully calibrated instrument responses of the SPITS seismometers also provides a significant benefit over typical temporary geophone deployments having a similar arrangement when it comes to MFP.

4. From experience (and this comment is a bit out of curiosity), there is typically some source smearing for events outside the array such that the distance of the sources cannot be well constrained. I would expect this to happen also for the class II events, but it does not seem to be the case. Also from Fig. 4F, the distance seems to be quite well constrained. Can you comment on that?

We observed much improved source range constraint using coherent-MFP than incoherent-MFP (see revised version of Figure 4 given later in this response). In addition, some MFP implementations only use phase information, but we found that the variation of amplitudes across the array gives important additional constraint when estimating the source position. For example, two sources positioned along a common azimuth located well outside the array will have very similar relative arrival times across the array (which is why distance is relatively poorly constrained by only considering wave phase). However, the furthest source will have less variation in amplitude across the array because the array aperture is a smaller fraction of the total propagation distance. The closer source will produce a larger contrast in amplitude between the nearest and furthest array element because the array aperture is a larger fraction of the total propagation distance. The contrast in amplitudes across the array is therefore a key constraint on the source range, in particular.

In this study, we benefit substantially from the fact that the SPITS seismometers are carefully calibrated. Inconsistent ground coupling and/or poorly constrained instrument responses both work to reduce the reliability of amplitude information for temporary seismic deployments (such as Romeyn et al. 2021) where it then becomes necessary to normalize amplitudes or discard them entirely and only consider the signal phase in MFP, such as in Walter et al. (2015), Chmiel et al. (2016), etc. The relatively large interstation distances at SPITS (aperture is ~1 km) are also beneficial in this specific case, because amplitudes attenuate quite appreciably across the array for the local (<~10 km) events that we studied (so that contrast in relative amplitudes is quite well resolved).

The accuracy of the range to sources far from the array depends largely on how well the assumed geometrical spreading model represents the true amplitude variation across the array. As noted in point 2, the fact that the seismic events associated with mining activities at Gruve 7 show reasonable

correspondence with the location of the underground mine workings is encouraging and indicates that the geometrical spreading model is a reasonable approximation of reality.

*Romeyn, R., Hanssen, A., Ruud, B. O., Stemland, H. M., and Johansen, T. A.: Passive seismic recording of cryoseisms in Adventdalen, Svalbard, The Cryosphere, 15, 283–302, https://doi.org/10.5194/tc-15-283-2021, 2021.*

*Chmiel, M., Roux, P., and Bardainne, T.: Extraction of phase and group velocities from ambient surface noise in a patch-array configuration, Geophysics, 81, KS231-KS240, 2016.*

*Walter, F., Roux, P., Roeoesli, C., Lecointre, A., Kilb, D., and Roux, P.-F.: Using glacier seismicity for phase velocity measurements and Green's function retrieval, Geophysical Journal International, 201, 1722-1737, 2015.*

Terminology: The wording could be more consistent. The text jumps around between e.g. ice wedge and segregation ice or frost quake and cryoseism. If it is not the same that is meant, please further specify each of the concepts.

This is a good point and was also brought up, in particular, in RC4. As we discussed in the author response to RC4, we suggest adding a conceptual model and revising the associated terminology to make it clearer what processes have been interpreted.

- Thermal-stress model: It is mentioned, that ignoring some of the temperature dependences (lines 244-248) results in different model formulations, that other studies used previously. Why do you chose this specific model and how would your results be affected by e.g. using the model proposed by Mellon (1997), or Podolskiy et al. (2019)?

These models can be considered a closely related family and we wanted to show clearly how they are related to one another. The simplifying assumptions made by Mellon (1997) and Podolskiy et al. (2019) were completely appropriate to those studies. Primarily, our formulation provides a convenient basis to compare those models. If we implemented the Mellon (1997) or Podolskiy et al. (2019) assumptions the modelled thermal stress would be broadly similar. However, when attempting to correlate the observed and predicted number of frost quakes at the level of detail of e.g. Figure 10, the second order contributions to thermal stress become relevant. It is important to note that the previous studies model the ground temperature profile, while we use a detailed record of in-situ borehole ground temperature measurements. This is perhaps the most significant factor that allows us to delve a little further into the fine detail of second order terms in the thermal stress model than was appropriate for, e.g., the Mellon (1997) and Podolskiy et al. (2019) studies.

**Line-specific comments**

Line 27: pressure release → stress release?

Yes we agree, stress release is more accurate terminology. This will be changed in the revised manuscript.

Lines 39-40: "These structures form …". I am having trouble to understand this process, maybe consider rewriting this sentence.

The purpose is to communicate the formation of the wedge structures that lead to the surface expression of ice-wedge or sand-wedge polygons. The development of these wedge structures is well documented in the literature but we see the need to improve the clarity of the text and suggest changing to the following:

> "Ice-wedge or sand-wedge polygons are a widely observed geomorphic feature in the periglacial environment (e.g. Black, 1976; Matsuoka et al., 2004). These wedges form when water that infiltrates and freezes to ice, or wind transported sand grains, hold open an initial thermal contraction crack that subsequently becomes an enduring plane of structural weakness (Lachenbruch, 1962; Mackay, 1984; Matsuoka et al., 2004; Sørbel and Tolgensbakk, 2002)."

Lines 49-57: What's the difference between ice wedges and segregation ice. As far as I understand one can broadly distinguish them as vertical and horizontal ice structures in the subsurface, respectively? Consider to add some definition here, if applicable.

Good point. We have addressed this more extensively in the author response to RC4, where we suggest adding a conceptual model and description of the corresponding physical processes. We agree that this wasn't covered in enough detail in the initial manuscript.

Line 61: "… InSAR has used …" → has been used

Thanks for the catch, this will be changed in the revised manuscript.

line 73: "This study was motivated" → is motivated

Perhaps this is simply a stylistic preference, but past tense seems appropriate in this context given that we explain the initial study hypothesis development which precipitated the study.

line 73: sporadic? From the paper it seems there are quite many of these events?

Sporadic in the sense that the events occur at irregular intervals, in total they add up to a large population, but they are not recorded all the time or at regular intervals. We suggest that an appropriate substitute that might improve readability in this context would be "intermittent".

Line 96: Maybe add a reference after matched field processing, that describes the "broadband, coherent" approach? Because that's the special part in this study, right?

Good point, a reference to Michalopoulou (1998) would not be out of place here. On reflection, it would be better to be more general here e.g. "well suited to source localization using matched field processing" since this section is just introducing the data. The specifics of the selected matched field processing approach can then be introduced and developed in the methods section. Coherent-MFP is not really the special part of the study *per se*, since it was developed quite some time ago for ocean acoustics. However, we did find it to be a very suitable tool for this study and it is nice to be able to highlight that a technique developed within ocean acoustics can also find useful application in other fields.

Line 127: So the weather station is measuring the air temperature plus the temperature of the ground in 0.1m depth? Please clarify.

Yes, exactly. We suggest the following minor adjustment to make this clearer:

"… the Janssonhaugen Vest weather station (see Figure 1), which was installed in September 2019, includes hourly sampled records of air temperature and ground temperature at 0.1 m depth. It therefore provides a basis to compare depth and temporal sampling effects against the longer duration, more coarsely sampled P11 record."

Line 131: What do you mean by "first-pass"?

We mean that the detection is a coarse initial step in a signal identification/classification procedure that contains multiple steps (e.g. the coal mine events are subsequently separated out based on inferred source location). We fully agree that the readability can be improved here and suggest the following revised text:

"The purpose of the event detector is to make an initial, coarse, automatic identification of short duration seismic signals, which should be distinguished from both background noise and longer duration local and regional seismic events…"

Lines 137 and following: A bit difficult to follow here – for the STA you take the envelopes and smooth them with a 1s sliding windows and for the LTA you smooth this curve once more with a 20s sliding window? Please clarify and maybe rewrite the text.

Yes, this is correct. We will clarify this in the revised manuscript.

Line 180: I think it should be the absolute value of the term after the sum. As is, it would be a complex MFP amplitude. Same for equation (7). Please check.

No, there shall not be an absolute value sign in Eqs. (4) and (7). These matrix operations are quadratic forms that guarantee a non-negative real valued output. Note that the original papers from ocean acoustics that introduced the MFP (and their variations) got it right (e.g. Michalopoulou et al. 1998), but that the (unnecessary) absolute value sign has somehow later found its way into the geophysics literature.  We suggest to add the sentence:

"The matrix operations in Eq. (4) are quadratic forms that formally guarantee a non-negative real-valued output."

Line 254-256: So you basically do a forward modeling using the measured temperature time series at a certain depth (and the parameters from Table 1) to calculate the resulting stress at this depth? If so, maybe strengthen this point here.

Yes, this is correct. We agree that this point can presented more clearly and suggest we replace the sentence

"We assume the stress at a given depth is decoupled from the stress at adjacent depths (Mellon, 1997) and solve Eq. (12) for the temperature timeseries at the selected depth of investigation."

with

"We solve the forward model Eq. (12) numerically to obtain a time series of the resulting horizontal stress $\sigma(z,t)$ at depth z. The crucial input to the forward model is the measured temperature time series $T(z,t)$ at depth z."

Correspondingly, we will add a short statement at Line 224 that we assume the vertical stress is zero, in line with Mellon (1997), corresponding to the assumption that the overburden is negligible in the shallow subsurface.

Line 280: So only less than 100 events were recorded by less than five stations and thus discarded?

Yes, that is correct. It makes sense to apply this check as part of a general procedure, because we were aware that instrument downtime due to maintenance or malfunction is a potential issue. It was therefore encouraging and a testament to the ongoing planned maintenance conducted by NORSAR that nearly all of the detected events were recorded by at least five seismometers. For the purpose of transparency, it is important that this is communicated, even though the number of discarded events was small.

Line 293: "are" is missing before "associated"

The "are" comes a few words earlier because of the chosen sentence formulation:

"Using coherent MFP, we find not only are these events associated with more distant sources (Figure 4f), they also have a consistent azimuth."

But we can happily rephrase this sentence to improve readability:

"Using coherent MFP, we find that these events are associated with more distal inferred source positions (Figure 4f) that also have a consistent azimuth."

Line 294: I see that compared to your previous study, nine seismic stations can be considered an array that coarsely samples the spatial domain, but I think this cannot be considered a general statement. Maybe relate this to your previous study.

It is clear that for these data, the SPITS array coarsely samples the spatial domain, as was commented on in major comment number three. The general statement that coherent MFP is most beneficial for arrays that coarsely sample the spatial domain (when compared with incoherent MFP) was made by Michalopoulou (1998). We see that our data support Michalopoulou's findings and that is what we wish to communicate with this sentence. We suggest the following modification to make this clearer:

"coherent MFP decreases source localisation ambiguity for arrays that sample the spatial domain coarsely for a given range of observed wavelengths when compared to the incoherent scheme (consistent with Michalopoulou, 1998)."

Line 319: delete "due", same in line 326.

Sure, we will remove these in the revised manuscript.

Line 341 and following: This relates to a previous comment: To calculate the stress at a certain depth, does only a single temperature time series from this particular depth enter the calculation, or does it also include the vertical temperature gradient? What does the word "combination" in line342 imply?

"Combination" highlights that the measured ground temperature field consists of two datasets, one from the weather station (0.1 m record) and one from the borehole (0.2-15 m).

The second part of the question is answered in the response to the previous comment on Line 254-256.

Line 353: "Figure 9 …" → Figure 9a. It would also be interesting to show the event rate (e.g. events/day) as a line together with the calculated stress in Fig. 9b.

In order to give the event rate, it is necessary to apply histogram binning. We prefer to represent the event rate histograms as bar rather than line plots. We will clarify in the figure and caption that the black bars in Figure 9b show the event rate binned to a 6-hour interval.

**Figures**

Figure 2: It took me a while to understand what's actually shown, since this is a continuous time series split into several subfigures. I would either merge the graphs of each row and/or write the year as text into the graphs, to make it easier for the reader.

This is a good point and we suggest the following revision to improve the readability of the figure. The splitting of the continuous timeseries is now mentioned explicitly in the figure caption to improve clarity.

[Figure]

**Figure 2 – Illustration of spatiotemporal borehole temperature recorded at the PACE P11 borehole on Janssonhaugen. A long-term warming trend is observed below the active layer that is subject to seasonal freeze-thaw. The continuous timeline is split across multiple figure panels.**

Figure 3: a) and b) are missing, but are referenced in the text. Also, in the caption, please provide more detail on what is actually shown.

Thanks for picking up on this oversight. We suggest the following revised figure and caption to address this (which also addresses comments from RC2):

[Figure]

**Figure 3 – Example of (a) timeseries of vertical component ground motion and (b) event detection using the STA/LTA (short term/long term average) detector. Short duration events with sufficient amplitude and array coherence are selected while longer events such as the high amplitude example at 04:50 are ignored. Inset boxes show detailed views for a specific detection.**

Fig 4: The crosses of the stations are hardly visible in subplots g, h, i

Thanks for the feedback. We suggest the following modified figure to improve readability:

[Figure]

Figure 4 – (a), (b) Examples of Class I events with significant amplitude variation across the array and relatively close source position inferred by (d), (e) coherent MFP. (c) Example of a Class II event with little amplitude variation across the array and a more distal (f) coherent MFP inferred source position. (g), (h) & (i) show corresponding incoherent MFP results, Eq. (4), demonstrating the improvement gained by coherent MFP. Station numbers in (a), (b) & (c) correspond to the labels annotated on the MFP panels where the seismometer locations are marked with black crosses.

Fig 6: The seasonality is hard to see from the figure. I suggest to give the total number of events shown in each panel e.g. in their titles.

Thanks for the suggestion, we'll add this to the revised manuscript. We will also add the total number of events shown in each figure panel to Figure 5 for consistency.

Fig 7b-d: Being non-trained in this, it is difficult for me to spot the boulder producing scarps and solifluction lobes. Consider adding annotations to the images.

This is useful feedback. There is a fine line between adding annotations and obscuring the details in the images that are interpreted. Ideally one could include both fully annotated and unadorned figures, but in the interests of brevity we suggest the following revised figure to retain some of the orthophotographic details while adding a sufficient level of annotation to guide the reader's eye:

[Figure]

Figure 7 – (a) Dec-Feb events as plotted in Figure 6 with orthophotograph details illustrating the geomorphologic features associated with the most seismically active areas. Erosional scarps (b), (d) are annotated in red and associated boulder fields in yellow, and frozen debris/solifluction lobes (c) are annotated in blue. A faintly visible area of polygonal patterned ground is annotated in green in (c). Orthophoto © Norwegian Polar Institute (npolar.no).

Fig 10: Maybe it would be better to show the event detection rate as a line instead of the vertical bars? What is the apparent stress? Please specify.

The event detection rate is shown as a frequency histogram in Figure 10b. In Figure 10a the colored bars help to avoid clutter. It is a good point that we could improve the labelling to make it clearer that the detection rate shown by coloured bars in Figure 10a is essentially the same as the "Array local events" in Figure 10b. We suggest labelling both as "Event Class I detections" to improve this.

Apparent stress is the modelled thermal stress corrected by dissipation due to tensile cracking. It is a good point that this should be more clearly defined in section 3.3.1 (Line 264-272) and a cross reference could usefully be included in the figure caption to guide the reader. Upon reflection, "pre-fracture stress" might be more descriptive than "potential stress" and "post-fracture stress" would make a good substitute for "apparent stress".

Fig 11: I think it would be instructive to show only the class I events and again maybe as a line or as bars. You have shown that earlier, that class II events are independent of the thermal stress, so it would be better to focus on your finding that the closeby events are related to the stress.

This is good feedback. We have considered this possibility, but in order to show the class I events as lines or bars representing frequency, we would need to apply histogram binning. This is what we

show in Figure 12, where we make a direct comparison to the modelled number of frost quakes accounting for tensile strength and stress release. We also think that it is quite useful that Figure 11 gives an overview of the results of the study using data in the rawest form practical.

---

## Author Comment (AC2)

The authors of the manuscript "Long term analysis of cryoseismic events and associated ground thermal stress in Adventdalen, Svalbard" performed a study on a large temperature and seismic database of Adventdalen valley on the island of Spitsbergen, gathered in the past two decades. The seismic data are then evaluated using STA/LTA and MFP approaches to a) filter out cryoseismic events and distinguish them from the mine activities and b) figure out the activity source location. The spatiotemporal temperature data are used to compute the stress history at different depths of the ice layer. Elastic, thermal and viscous strains drive the stress calculations. A simple fracture model is used to predict the possible cracks and cryoseismic events and compare them to the recorded seismic data. The authors concluded that there is good agreement between the model predictions and the recorded data.

In my opinion, the current manuscript lacks enough novelty and depth to get published in The Cryosphere journal. I do not have enough expertise to judge the MFP calculations section, but I hope the comments I made for the thermal stress and fracture sections help the authors to elevate the existing manuscript to The Cryosphere journal-level quality.

We disagree that the manuscript lacks novelty. The relatively recent publications by Okkonen et al. (2020) and Podolskiy et al. (2019) are perhaps the closest in scope and were an important inspiration for this study. However, the present manuscript diverges in numerous significant aspects from these previous studies, particularly with respect to the use of a measured rather than modelled ground temperature record and the use of a large catalogue of thousands of individually detected and located events spanning many years. The result of these fundamental differences is that the degree of overlap with previous studies is quite small.

We do appreciate the feedback and have used the review comments as a basis to identify a number of improvements that can be made to the revised manuscript.

The Introduction is not coherent. I could not find a clear bridge between paragraphs, and also the relation between written paragraphs and the paper's goal is not clear to me.

We will revise the introduction to improve the bridge between paragraphs in the revised manuscript.

Figure 2 needs more description. I assume each sub-plot corresponds to a certain year; you need to show that in the figure or caption.

In response to this and a similar comment from RC1 we suggest replacing Figure 2 with the following updated version.

[Figure]

**Figure 2 – Illustration of spatiotemporal borehole temperature recorded at the PACE P11 borehole on Janssonhaugen. A long-term warming trend is observed below the active layer that is subject to seasonal freeze-thaw. The continuous timeline is split across multiple figure panels.**

Figure 3: It would be nice if you zoom in into one of the detected events for better clarity of your method.

We suggest including the following updated version of Figure 3 in the revised manuscript.

[Figure]

**Figure 3 – Example of (a) timeseries of vertical component ground motion and (b) event detection using the STA/LTA (short term/long term average) detector. Short duration events with sufficient amplitude and array coherence are selected while longer events such as the high amplitude example at 04:50 are ignored. Inset boxes show detailed views for a specific detection.**

Figure 6: It is hard to distinguish differences between seasons only by checking these contours. Adding numbers to either image or in the caption would help readers to notice the fluctuations across seasons.

This is a good suggestion; we can add the number of events corresponding to each subfigure. We will also do the same for Figure 5 for consistency.

[Figure]

**Figure 6 – Detail view of Janssonhaugen overlaid with spatially binned (50×50 m) distribution of coherent MFP inferred seismic source positions plotted by season for events recorded between August 2004 and July 2021. Positions of SPITS seismometers are indicated by white crosses. Orthophoto © Norwegian Polar Institute (npolar.no).**

Page 16, 325: Your justification here to exclude summer-autumn events from your study does not seem sufficient to me. I am looking for better justification in the rest of your paper…

This seems to be related to a misunderstanding based on a poor choice of words on our part. To clarify, summer-autumn events are not excluded from the study. Note that figures 5, 6, 9, 10, 11 & 12 all include summer and autumn seasons. A specific, local, spatial domain is excluded from the spatially delineated Class I event cluster. This domain corresponds to a river valley that is particularly active during the summer-autumn. InSAR results have also confirmed that this area is highly dynamic in the summer (as cited in the manuscript). This activity is likely related to processes of fluvial erosion and river bank oversteepening, rock glacier movement etc., which do not belong to the same class of events as those which we interpret as cryoseisms. Put another way, event Class I was isolated using a simple radial distance cut-off of 1500 m from the centroid array, excluding the river valley south of the array that overlaps this zone (where another,  minor event class resulting from different dynamic processes dominates).

We agree that including the phrase "summer-autumn" to describe a spatial cluster, was unfortunately rather ambiguous. To improve clarity, we suggest rephrasing from:

> "By selecting the subset of events with inferred source positions within ~1500 m of the array centroid and excluding the cluster of summer-autumn events south of the array, we isolated a total of 42,432 class I events recorded between July 2004 and July 2021."

To the following:

> "By selecting the subset of events with inferred source positions within ~1500 m of the array centroid, excluding the river valley/rock glacier area south of the array, we isolated a total of 42,432 class I events recorded between July 2004 and July 2021."

Page 16, 330: Again, the justifications in this paragraph are not enough and lack scientific statements. At least, I as a reader, expect to know what type of data you need to draw a more accurate conclusion.

No direct conclusion is to be drawn here; we are simply delineating the spatial extent of the cluster of seismic events that we categorise as event Class I and the broad seasonality associated with this cluster. In response to RC4 we will add a more detailed interpretation of the anomalous seismicity of the three identified areas so that the following sentence, that this comment relates to, will be removed from the revised manuscript:

"These areas may be associated with enhanced ground heat loss, thin or absent snow cover or elevated ground moisture/ice content (e.g. Abolt et al., 2018; Matsuoka, 2008), though we lack the field observations necessary to support this explanation for the anomalous seismicity of these areas."

Figure 8: I suggest reducing the legend of the plot to -0.5-1.5 for better contrast. I do not see values below -0.25 in the contour plots.

The observation that the values mostly lie within the range $-0.5 \times 10^7$ Pa to $1.5 \times 10^7$ Pa is correct. However, for readability it is very convenient that zero stress is white. One can observe that we have assigned a range of colours to the positive range of stress, while the values below ~$-0.25 \times 10^7$ Pa are uniformly black (so figure contrast will not be affected by the suggested change). It is desirable to convey that the magnitude of tensile stresses associated with thermal contraction during periods of cooling far exceed the magnitude of stresses associated with thermal expansion. The included colour scaling also makes clear this asymmetry.

Section 4.2: What are the initial and boundary conditions for solving equation 12?

This is a first-order differential equation with respect to the time variable, so we only need an initial condition. There are no boundary conditions for first-order temporal models. The initial condition is stated on line 242 of the manuscript:

"In order to solve Eq. (12) for $\sigma(z,t)$, we specify the initial condition $\sigma_0(z) = \sigma(z,t = 0) = 0$."

Page 17, 345: I do not understand how you associated the 20-30cm regolith to the peak stress in the ice above it. How the peak stress in the ice could lead to high stress in the rocks beneath it?

In the borehole we have temperature measurements from sensors installed at 0.2, 0.4, 0.8, 1.2, 1.6, 2, 2.5, 3, 3.5, 4, 5, 7, 10, 13 & 15 m depths. We also have a record from 0.1 m depth at the Janssonhaugen Vest meteorological station. Of all of these temperature records, the largest thermal stress is associated with the 0.2 m deep temperature record (as shown in Figure 9-b). Cracking is most likely to occur where the stress is highest. The regolith layer at Janssonhaugen is 20-30 cm thick. This gives an indication that thermal stress weathering/cryoturbation may be an important control on regolith depth when allowed to act over a long time. Aggradation of weathering material would be another explanation, but this less likely the case since the top of Janssonhaugen is a mountainous plateau where erosion is expected to dominate over deposition. If we had modelled the largest thermal stresses at a depth of 1 m, for example, we would expect the ground to be

heavily fractured to this depth and that over thousands of years a 1 m regolith layer might be formed.

We can add the key details from this discussion to clarify this in the revised manuscript, but will not include the previous paragraph in its entirety for the sake of brevity.

Figure 9: I am interested to see the contribution of each strain portion (elastic, thermal, viscoelastic) into the total stress where ever you report the stress value (Figs 8-11).

These components are interconnected through a differential equation (Eq. (12)), so one cannot simply decompose the resulting total stress into separate components in an additive manner.

Page 20, 395: This paragraph suits better in the conclusion section.

We think it is important to point out that the spatial variability of the subsurface temperature field is not constrained by the borehole temperature measurements used in this study, as this paragraph discusses. However, we don't think this topic fits as a main conclusion of the study.

Section Conclusion: This section is better to be named Summary rather than Conclusion. To enrich your paper's conclusion section (which should be the most important section) I suggest discussing pros/cons of your thermal and MFP model, potential improvements of your work, and maybe possibilities to apply your model to other geographical locations…

It is a perhaps a stylistic choice, but we prefer a brief conclusion summing up the most important results of the study. The possibility to apply the study methodology to other geographic locations doesn't need to be stated explicitly, but we can add the detail that future calibration experiments using controlled sources in known locations would improve the utility of SPITS for MFP studies.

I am curious if you noticed any pattern in the recorded quakes for daytime versus night times (heating vs. cooling periods)?

Janssonhaugen is situated on Svalbard in the high Arctic. Here the polar night, where no shortwave solar radiation is received at the ground surface, lasts from around 1-Oct to 28-Feb each year. During the summer, the sun does not set between 19-Apr and 23-Aug and solar insolation received at the ground surface also depends on local factors like snow cover and topography. We certainly observe that frost quakes were more frequently recorded during the polar night than during the period of midnight sun. It is, however, impossible to generalize that day and night correspond to periods of heating and cooling in the high Arctic if one assumes the typical definition of day and night as representing ~12-hour phases in the diurnal cycle. Interestingly, the diurnal temperature range on Svalbard is actually greatest during the winter (Przybylak et al., 2014), despite the complete absence of solar irradiation. This is explained by the intensity of winter storms and the advection of warmth to the region, driven 95% by atmospheric circulation and 5% by oceanic circulation (e.g., Bednorz, 2011). The complexity of the surface energy budget in this region (e.g., Westermann et al., 2009), further reinforces a key strength and novelty of the manuscript, i.e., that we utilize measured ground temperatures rather than modelling ground temperatures based on measurements of air temperature. We will add some details about insolation and the importance of synoptic weather systems in driving temperature variation to the description of the study area in the revised manuscript.

Bednorz, E. (2011). Occurrence of winter air temperature extremes in Central Spitsbergen. *Theoretical and Applied Climatology*, *106*(3), 547-556.

Przybylak, R., Araźny, A., Nordli, Ø., Finkelnburg, R., Kejna, M., Budzik, T., ... & Rachlewicz, G. (2014). Spatial distribution of air temperature on Svalbard during 1 year with campaign measurements. *International Journal of Climatology*, *34*(14), 3702-3719.

Westermann, S., Lüers, J., Langer, M., Piel, K., & Boike, J. (2009). The annual surface energy budget of a high-arctic permafrost site on Svalbard, Norway. *The Cryosphere*, *3*(2), 245-263.

---

## Author Comment (AC3)

**1 General Comments**

This article aims to study the hypothesis that short-duration seismic events at SPITS can be due to the frost quakes induced by thermal contraction cracking. The main interest of the study lies in the identification of seismic events that correspond to frost quake initiated by thermal stress.

Two event classes were defined by authors and the second class of seismic event was concluded due to the underground mining operations. The first event was found to be active during winter (Dec-Feb) and spring (Mar-May) and inactive in the summer time.

The article is well written and the research is interesting. A contribution from this paper is that the authors used the ground temperature measured by a series of thermistors in the numerical analysis. However, the proof of the hypothesis (especially for the frost action induced seismic events) is still not convincing. In my opinion, the paper needs to be revised to convincingly explain how frost action can initiate the dynamic response. The following comments are also needed to be addressed.

We thank the reviewer for their comments. We agree that strengthening the highlighted aspects will improve the revised manuscript and provide detailed responses to the specific comments covering these issues below.

First, the authors should include a fuller explanation of the physical mechanism of frost quakes and how the frost action could induce dynamic responses.

Thanks for this feedback. We can agree that the physical cracking mechanism should be presented in greater detail. This was also commented on by reviewer 4. As also stated in the author response to RC4, we suggest adding **Error! Reference source not found.** and the following description to more clearly illustrate the investigated cracking mechanism.

> Polygonally patterned ground indicative of cryoturbation of the active layer and/or ice wedges in the underlying permafrost are observed extensively across Janssonhaugen, except where downslope mass movements destroy or interrupt the formation of polygonal networks (Sørbel and Tolgensbakk, 2002). In this study, we observed anomalously high cryoseismicity associated with erosional scarps and a frozen debris/solifluction lobe, which are all associated with downslope mass wasting. We suggest that these downslope mass movements initiate or precondition transverse cracks or fissures to open (e.g. Darrow et al., 2016, Price, 1974). Water from rain or snowmelt infiltrates these fissures and freezes when temperatures drop below freezing (e.g. Darrow et al., 2016). Rapidly falling ground temperatures during winter then cause the surrounding ground to thermally contract. This causes the vein ice filling the frozen fissures to crack under tension, since the tensile strength of ice is lower than the surrounding ground and therefore constitutes a plane of weakness. Cracking relieves the accumulated thermal stress.

> This mechanism is analogous to the Lachenbruch (1962) model of thermal contraction cracking of ice wedges in permafrost. However, the cracking may occur more frequently or under milder surface cooling because 1) the frozen fissures may be pre-stressed by downslope gravitational forces making them more prone to failure and 2) the fissures extend to the ground surface where thermal stresses are largest. For the case of ice wedges, the ice wedge is located below the permafrost table (**Error! Reference source not found.**-c). There may be vein ice extending through the active layer only if the previous seasons thermal contraction crack has remained open through the summer thaw season. If the crack has closed so that vein ice is not formed during the early freezing season, initiation of thermal

contraction cracking of the ice wedge would require accumulation of thermal contraction stress at the level of the permafrost table, requiring a longer or more extreme surface cooling episode. Tensile cracks could also form in the active layer where ice veins/ice wedges are absent (as in **Error! Reference source not found.**-a), but this would require thermal contraction stresses exceeding the tensile strength of frozen ground, which is greater than that of polycrystalline ice and may therefore occur less frequently.

[Figure]

*Figure I - (a-d) Lachenbruch (1962) model of ice wedge formation. (a) Thermal contraction of frozen active layer initiates a tensional crack that penetrates into permafrost, (b) meltwater infiltrates and refreezes during the thaw season, (c) ice with lower tensile strength than surrounding ground forms a plane of weakness and cracks repeatedly over many years, (d) the crack-infilling cycle causes ice wedge growth and ground surface deformation that organizes into ice wedge polygons in 2D plan view. Mass movements such as, (e) erosional scarp and (f) frozen debris/solifluction flow initiate transverse cracks/fissures which become infiltrated by water that freezes to ice. (g) Thermal contraction of the surrounding ground and downslope gravitational stress causes the ice to crack under tension.*

Second, the classification of two classes of seismic events must be addressed more quantitatively, in order to classify and detect frost action related seismic events.

We have addressed why short duration signals are studied in response to several specific comments from the reviewer below. The two classes of events are ultimately distinguished based on their spatial distribution; those events located in the vicinity of the underground mine workings are assumed to originate from human activity in the mine.

Third, the mismatch between the wavelength of seismic events (could be more than 115 m) and the studying depth (top 15) needs to be better explained or addressed.

We move the specific comment to Line 299 here since it covers the same issue: Authors stated 'The mean MFP inferred propagation velocities for Class I events was 1150 m/s with a standard deviation

of 1100 m/s, indicating that they are dominated by surface waves. The large standard deviation may indicate the surface waves are dispersive with different frequencies propagating at different phase velocities.' In line 132, the authors also mentioned the signal is filtered to a range of 2.5-20 Hz. This gives us a wavelength around 115 m (1150 m/s divided by 10 Hz, average frequency) and the investigation depth in this paper is only about the first 15 m. The authors should explain this mismatch.

It is difficult to understand exactly what mismatch is referred to. We present the ground temperature field measured to a depth of 15 m, that suggests a possible seismic source via thermal contraction cracking at a depth around 0.2 m. We also present seismic records of surface ground motion and show that the spatial and temporal distribution of a specific subset of seismic signals is consistent with a thermal contraction cracking source. A crack/rupture of small size can produce a seismic surface wave signal with long wavelength. Frequencies down to 3 Hz are not-uncommon in sledgehammer surveys for MASW and these are perhaps even less energetic seismic sources. There is therefore no contradiction in associating signals with relatively long wavelength with a small magnitude, shallow seismic source.

It is likely that the thermal contraction cracks also excite body waves of higher frequency. The reason we don't observe these might be that they are above the Nyquist frequency (40 Hz) and/or that attenuation is too large. However, surface waves can be intuitively explained as constructive interference of body waves trapped near the surface and the resulting horizontal wavelength can be much longer than the wavelength of the dominant frequency (corner frequency in source spectrum of ground displacement) of the body waves excited by a small magnitude seismic source.

We must emphasize that we have not conducted a MASW study where we try to estimate the shear-wave velocity depth profile of the uppermost 15 m. If this were the case, the long wavelength of the recorded seismic signals would likely be problematic.

**2 Specific Comments**

Line 32: I would add a few sentences to explain why ground cooling induces cracking. Is it because of the volumetric expansion during the phase change from water to ice? Or it can be due to the formation of ice lenses which is associated with the water migration during the freezing process? Can also thawing contribute to cracking?

Thanks for this feedback. It is a good point that it would give a more complete picture to introduce both the thermal contraction and segregation ice mechanisms of cracking in this section. We suggest the following formulation:

> "Frost quakes are typically observed in association with rapid air and ground cooling, in the absence of an insulating snow layer and where sufficient moisture is present for ice to form (Barosh, 2000; Battaglia et al., 2016; Matsuoka et al., 2018; Nikonov, 2010). The source of frost quakes are cracks that may be initiated by different mechanisms including thermal contraction exceeding tensile strength (e.g. Lachenbruch 1962) or by the growth of segregation ice driven by the capillary migration of water to a freezing front (e.g. Walder and Hallet 1985, Peppin and Style, 2013). In this study we focus on the thermal contraction cracking mechanism, which is consistent with the association of transient ground acceleration events with rapid cooling episodes and cold winter temperatures reported by

Matsuoka et al., (2018) and consistent with previous descriptions of frost quakes (Barosh, 2000; Battaglia et al., 2016; Nikonov, 2010; Okkonen et al. 2020)."

We are not aware of a mechanism by which thawing can directly contribute to cracking, to our knowledge desiccation cracking would be the closest. However, evidence that soil desiccation cracks produce measurable acoustic emissions or excite seismic waves seems to be limited.

Line 34: Frost heave is an upward swelling of soil due to an increasing presence of ice as it grows towards the surface (continuously delivers water to the freezing front via capillary action). I am having difficulty understanding that frost heave is ʾrapidʾ and also ʾelasticʾ deformation. The frost heave requires delivering unfrozen water contentiously to the freezing front via capillary force, which is likely a slow process (Darcyʾs Law). It is not necessarily elastic either given its large deformation.

Good point. The term "frost heave" was poorly placed in this sentence, which could be better formulated as:

"Cryoturbation can be understood as a combination of slow creep (frost heave, e.g., Rempel, 2010) and rapid elastic cracking (frost quake) deformation of frozen ground and causes damage to roads requiring billions of dollars annually to repair in the United States alone (DiMillio, 1999)."

Line 49: What is the mechanism for the segregation ice growth in bedrock? If it is the same as the frost heave, where does the capillary force come from?

We suggest changing the order of sentences and to split up the references to previous studies, which should make this clearer. We propose the revised formulation:

"Frost cracking driven by segregation ice growth is also an important agent of bedrock erosion in cold mountainous areas, where rockfall, active screes and high headwall erosion rates are observed in areas where frost erosion is most intense (Hales and Roering, 2009; Hales and Roering, 2007; Scherler, 2014). An important mode of crack growth in water permeable bedrock is the migration of water to form segregation ice bodies (Hallet et al., 1991; Murton et al., 2006; Walder and Hallet, 1985) that is similar to the mechanism by which ice lenses develop in freezing soil (Peppin and Style, 2013). Segregation ice growth, frost heaving and creep on slopes leads to the development of solifluction lobes and sheets (Cable et al., 2018; Matsuoka, 2001). Solifluction is broadly defined as the slow mass wasting resulting from freeze-thaw action in fine-textured soils (French, 2017; Matsuoka, 2001) and occurs due to the asymmetry between frost heaving perpendicular with the sloped ground surface and vertical subsidence upon thawing under the force of gravity."

Line 119: It would be useful to also provide the temperature distribution in the permafrost at the location of P11 (even in the supplementary information).

Since the active layer is ~2 m thick, the interval from 2 m to 15 m is permafrost. Perhaps a misunderstanding arose around the sentence (Line 123-125):

"The ~2 m thick active layer (Christiansen et al., 2020) is sampled by thermistors at 0.2, 0.4, 0.8, 1.2, 1.6 and 2 m and there is significant inter-annual variability in the magnitude of summer warming and winter cooling."

For clarity we could add a following sentence:

"The upper part of the permafrost at the P11 borehole location is sampled by thermistors installed at 2.5, 3, 3.5, 4, 5, 7, 10, 13 and 15 m."

The temperature distribution in the uppermost permafrost is therefore already shown in Figure 2. The permafrost deeper than 15 m is not considered relevant to this study because temperature changes occur over long inter-annual timescales (the reader is referred to Isaksen et al. 2001, who analyze the deeper interval). As shown in Figure 8 the thermal stresses are already quite small below ~5 m.

Line 131: Authors need to provide more explanation to illustrate why cryoseismic events have a shorter duration than other seismic events.

In this section of the manuscript, the events have not been interpreted as cryoseisms, we are just describing a method to isolate short duration seismic events. The reviewer's point remains valid though, and we can add a sentence to the introductory paragraph where the study hypothesis is outlined (Line 74) explaining this.

The main reason why cryoseismic events have a shorter duration than other seismic events is that the source is much closer to the array. The different propagation velocities of P, S and surface waves (dispersion) mean that the larger the source distance becomes, the more spread out the wavefield becomes and the longer the ground motion duration will be. We can add a sentence or two to the revised manuscript to address this point.

A secondary point is that duration of ground motion can be used as a metric of earthquake magnitude (e.g. Lee et al., 1972; Mousavi and Beroza, 2020; Tsumura, 1967). It is also well established that magnitude correlates with the area of rupture, length of surface rupture etc. (e.g. Wells and Coppersmith, 1994). Shorter duration seismic events can therefore also be indicative of rupture along smaller cracks, with less energy release than larger magnitude tectonic earthquakes. Such small magnitude events are only recordable locally because the amplitude of ground motion decays below the background noise level over longer distances. Romeyn et al. (2021) showed that cryoseismic events could also be recorded in Adventdalen ~10 km from Janssonhaugen. We were not able to detect these same events at SPITS (presumably because the seismic waves had attenuated below the background noise level at this range), though the same ground cooling episodes were associated with similar clusters of events with estimated source positions in closer proximity to Janssonhaugen.

Line 136: It would be useful providing any physical interpretations of the ratio of short-time averaged amplitude and long-time-averaged amplitude (explain why this ratio can be used to detect cryoseismic events). I would also add a few sentences to explain why choose Hilbert transform for the short-time-averaged amplitude and moving average for long-time-averaged amplitude.

The STA/LTA method is simply a way to automatically identify higher-amplitude transient signals embedded in background noise and has been commonly used for decades in seismology. Thermal contraction cracking rapidly releases accumulated elastic stress, exciting transient seismic waves (the

energy release is not continuous) that are recorded as transient signals of ground motion. However, the transient signals detected by an STA/LTA detector could have any origin so it would not be appropriate to interpret the detections in isolation. We do filter transient signals according to their duration within this procedure, which as we have responded elsewhere is because longer duration signals correspond to longer rupture, higher magnitude, more distal earthquake sources. Importantly, the STA/LTA detections also included seismic events associated with mining activities. We were able to interpret a specific category of detections as cryoseismic events only in combination with MFP to estimate their source positions and temporal correspondence with periods of rapid cooling and increased thermal stress.

The Hilbert transform is only used to calculate the trace envelope and this is also common practice in signal processing.

To improve this section, we suggest rephrasing to:

> "Events are detected based on anomalous values of short-time-averaged (STA) amplitude divided by long-time-averaged amplitude (LTA), i.e., the classic STA/LTA approach widely used in seismology (e.g. Allen, 1982; Trnkoczy, 2009). The STA window length should be comparable to the target signal duration, while the LTA represents the background noise level. When the STA/LTA ratio exceeds a given threshold, an event is triggered. In our implementation, the STA is given by the one-second moving average smoothed trace envelope for each seismogram. The LTA is the STA further smoothed according to a 20 second period moving average."

Line 141: Authors stated that 'By visual inspection of test periods, we found that this emphasizes very local events with large amplitude variation across the array, while still ensuring that there is at least some coherency across the array'. It would be better to explicitly indicate what 'this' represents. More importantly, there should be a figure to show what authors captured by their visual inspection and prove how it emphasizes local seismic events.

We suggest re-phrasing to:

"By visual inspection of test periods, we found that the 80th percentile station-STA emphasizes short duration events with large amplitude variation across the array, while still ensuring that there is at least some coherency across the array (see Figure 3)."

[Figure]

**Figure 3** – Example of (a) timeseries of vertical component ground motion and (b) event detection using the STA/LTA (short term/long term average) detector. Short duration events with sufficient amplitude and array coherence are selected while longer events such as the high amplitude example at 04:50 are ignored. Inset boxes show detailed views for a specific detection.

Figure 3 is a good illustration of the kind of events that are selected. One may also observe that events with longer duration and high amplitude are not selected and neither are short duration events with low amplitude. An important case not shown in the figure is that we also manage to ignore spurious noise spikes that are only observed on a single seismometer. We can add a note covering single channel noise spikes at Line 153.

We take the point that saying "local events" in this section of the manuscript is perhaps a leap of logic and have modified that to "short duration events". As we have responded elsewhere, duration of ground motion can be used as a metric of earthquake magnitude (e.g. Lee et al., 1972; Mousavi and Beroza, 2020; Tsumura, 1967) and source distance. It is also well established that magnitude correlates with the area of rupture, length of surface rupture etc. (e.g. Wells and Coppersmith, 1994). Shorter duration seismic events are therefore indicative of rupture along smaller cracks, and less energy release than larger magnitude tectonic earthquake. Such small magnitude events are only recordable locally because the amplitude of ground motion decays below the background noise level over longer distances. Notably, we have also used MFP to prove that the short duration signals correspond to local seismic sources.

Line 151: Authors concluded that the ratio of short-time-averaged and long-time-averaged peaks must have amplitude larger or equal to 10 and occur at least 5 seconds apart from one another. Authors need to explain how they drew this conclusion, or clearly indicate this is an assumption if that is the case.

OK, we can give a bit of extra detail here and propose the revised formulation:

"The STA/LTA threshold was set to 10. Furthermore, after a trigger, no new events are declared within 5 seconds after the STA/LTA was exceeded to avoid detecting the same events multiple times. As for the STA and LTA lengths, these parameters were found to be appropriate for detection of short-duration signals coherent across the array, while avoiding false triggers (noise bursts at single stations), by visually inspecting test periods."

Line 209: It is common to add the unit for every applicable parameter defined in the manuscript.

We follow the custom from physics literature that units are not listed when deriving the physical model. Units for every applicable parameter are listed in Table 1 which is introduced when we discuss the parametrization and numerical solution to Eq. (12).

Line 253: Authors stated that 'A significant novelty of this study is that the ground temperature profile at Janssonhaugen has been logged by a series of thermistors installed in the 15 m deep P11 borehole at 6 hr intervals since April 1999'. How did the authors use these measurements in the analysis in the entire study domain (I suspect interpolation is required)?

Good point. We can add a note to Figure 2 and Figure 8 that the measured temperatures were interpolated to regular 10 cm depth intervals using a spline interpolant. In Figures 9, 10 and 11 the measured temperature timeseries are used without interpolation.

Line 254: Authors need to elaborate: 'We assume the stress at a given depth is decoupled from the stress at adjacent depths'. How exactly did the authors decouple the stress components in space since they are dependent on each other (if assuming the soil is continuum media).

We agree that "decoupled" is not the most suitable term. We applied the standard assumption of Mellon (1997) that the vertical stress is zero (because the overburden is assumed to be negligible in the shallow subsurface). As shown by Mellon (1997), this allows the horizontal stress at a given depth to be calculated independently, even though the stresses at different depths are directly connected through the temperature profile. We will state this more clearly around Line 224 and remove the sentence highlighted by the reviewer on Line 254.

Equation 10: Can this model be used to study the thawing of frozen soils?

Yes, the model is compatible with both thermal expansion and contraction. The extent to which it could be used to study thawing of frozen soils would depend on the specific aspect one wanted to study.

Table 1: How the tensile strength is calculated? Is it always 1 MPa? Should it also be temperature dependent?

Yes, it is always 1 MPa. While the compressive strength of polycrystalline ice has been shown to vary significantly with temperature, the tensile strength seems to be a weak function of temperature and is mostly controlled by grain size (e.g. Petrovic, 2003). We can add a reference to Petrovic (2003) to Table 1 to clarify this point. We suggest the following revision to the note in Table 1:

"Currier and Schulson (1982), varies according to grain size for randomly oriented polycrystalline ice (finer grained ice stronger), insensitive to temperature (Petrovic, 2003)."

Petrovic, J. J. (2003). Review Mechanical properties of ice and snow. Journal of Materials Science, 38(1), 1-6. 10.1023/A:1021134128038

Line 288: Authors stated that 'Event class I is characterized by significant amplitude variation and arrival time differences across the array seismometers'. A quantification method (e.g., L2 distance) is needed to describe the significant amplitude variation as well as similar amplitudes (line 291).

This sentence is a description of the first-order qualitative signal characteristics that the reader may observe in Figure 4. Importantly, the amplitude variation is dealt with quantitatively in the matched field processing method (ref. Eq. (1)) where it provides an important data constraint on the estimation of source position by assuming a model of amplitude attenuation by geometrical spreading.

Also, authors predicted the source of seismic events is around 1500 m. What is the uncertainty of this prediction?

We state that the Class I events "occur in relatively close proximity, within about 1500 m of the centroid of the array", meaning that the events we interpret as belonging to this class have estimated source positions that fall approximately within a circle with a radius of 1500 m, centered over the array. This is not a prediction, but a heuristic threshold necessary to delineate the cluster in space. To clarify this, we suggest rephrasing to "occur in relatively close proximity and are clustered inside a circle with radius of ~1500 m centered over the array".

Line 303: It is difficult to determine the dominant wave type based on merely the estimated propagation velocity. Could the 1150 m/s also correspond to body wave?

Good point, a direct or refracted vertically polarized shear wave propagating through the shallow subsurface at varying depths could in theory have a similar velocity. However, we might expect body waves to have higher frequencies. Surface waves are also expected to be excited for a near-surface source and then often dominate the signal. From the waveform example included below (Figure II) it is also clear that the highest amplitude is observed for a phase arriving after the P and S wave, i.e., most likely a Rayleigh wave.

To address the comment, we suggest re-phrasing from:

"The mean MFP inferred propagation velocities for Class I events was 1150 m/s with a standard deviation of 1100 m/s, indicating that they are dominated by surface waves."

To:

"The mean MFP inferred propagation velocities for Class I events was 1150 m/s with a standard deviation of 1100 m/s, implying a relatively shallow propagation path."

We could also include Figure II as an appendix showing illustrative three-component seismograms for the two event classes. There is some indication of phase rotation between the vertical and radial components for the frost quake (Figure II-a), which would indicate a Rayleigh wave. The Gruve 7 event that is interpreted to be dominated by P wave energy (Figure II-b) has vertical and radial components that are more similar to one another. We may also observe that the largest amplitudes for the Gruve 7 event correspond to the first arrival (P-wave), whereas the largest amplitudes for the frost quake arrive later (Rayleigh wave).

[Figure]

*Figure II – Three-component seismograms from SPITS station SPA0 for (a) an example of event Class I (frost quake) and (b) an example of event Class II (Gruve 7 mining activity). The signals are bandpass filtered to 2.5-35 Hz and the horizontal components of ground motion (measured in NS and EW orientations) were rotated to radial and transverse to the source bearings estimated by MFP (see Figure 4e and 4f).*

Figure 3: It is difficult to understand why longer seismic events have high amplitude. Authors might want to elaborate on the relation between the duration and the amplitude of displacement (or velocity and acceleration).

Similar to our response to an earlier comment, duration of ground motion can be used as a metric of earthquake magnitude because of stronger coda waves (e.g. Lee et al., 1972; Mousavi and Beroza, 2020; Tsumura, 1967). Since the magnitude is larger the amplitudes can be large, but this depends on the source-receiver distance and rate of attenuation. Therefore, high amplitude events are not

necessarily high-magnitude events and do not need to have a longer duration (e.g., frost quakes). More importantly, the larger the source distance, the more spread-out are the seismic signals (longer event duration) because of different propagation velocities of seismic phases. The detector is designed to separate short duration seismic signals from background noise and longer duration signals that *may* be high amplitude (Line 131). If the events of interest were always higher amplitude than all other events a simpler detection scheme based on amplitude thresholding would have been possible. To clarify why we choose to isolate only the short duration events, we suggest adding a sentence to the introductory paragraph outlining the study hypothesis (Line 74), which also covers the earlier comment relating to line 131.

**References**

Allen, R.: Automatic phase pickers: Their present use and future prospects, Bulletin of the Seismological Society of America, 72, S225-S242, 1982.

Lee, W. H. K., Bennett, R., and Meagher, K.: A method of estimating magnitude of local earthquakes from signal duration, Citeseer, 1972.

Mousavi, S. M. and Beroza, G. C.: A Machine-Learning Approach for Earthquake Magnitude Estimation, Geophysical Research Letters, 47, e2019GL085976, 2020.

Petrovic, J. J.: Review Mechanical properties of ice and snow, Journal of Materials Science, 38, 1-6, 2003.

Sørbel, L. and Tolgensbakk, J.: Ice-wedge polygons and solifluction in the Adventdalen area, Spitsbergen, Svalbard, Norsk Geografisk Tidsskrift-Norwegian Journal of Geography, 56, 62-66, 2002.

Trnkoczy, A.: Understanding and parameter setting of STA/LTA trigger algorithm. In: New Manual of Seismological Observatory Practice (NMSOP), Deutsches GeoForschungsZentrum GFZ, 2009.

Tsumura, K.: Determination of earthquake magnitude from total duration of oscillation, Bull. Earthq. Res. Inst., 45, 7-18, 1967.

Wells, D. L. and Coppersmith, K. J.: New empirical relationships among magnitude, rupture length, rupture width, rupture area, and surface displacement, Bulletin of the Seismological Society of America, 84, 974-1002, 1994.

---

## Author Comment (AC4)

Romeyn et al. presents novel and long time series of data on seismic events in a permafrost environment. The data analysis clearly distinguishes two kinds of events, derived from mining activities and natural sources (cryoseisms). The results are very interesting and worth publishing in The Cryosphere. In terms of periglacial geomorphology, however, I would suggest several corrections and clarifications, mainly on the terminology and data interpretation.

**Major comments**

1. Thermal contraction cracking and frost cracking appear to be confused (e.g., in Section 4.2): Please distinguish between frost cracking (which occurs when ground is 'freezing' and by a mechanism similar to frost heaving) and thermal contraction cracking (which occurs when 'frozen' ground is subjected to rapid cooling. Frost cracking associated with segregated ice tends to produce horizontal cracks which can result in rock fragmentation with repeated freeze-thaw cycles over thousands of years. In contrast, thermal contraction cracking produces vertical cracks with spacing of several meters and may not contribute to rock fragmentation (regolith formation).

This is a fair criticism. We have perhaps allowed the common usage of "frost polygons" to carry across to the description of cracking. We agree it is important that we consistently present that our model is suited to investigating thermal contraction cracking and will make sure this is clear in the revised manuscript. We suggest adding Figure i to more clearly illustrate the hypothesized cracking mechanism we aim to test (which also relates to some of the reviewer's other comments).

---

## Referee Report (RR1)

The authors' responses to my previous comments are acceptable. I believe the current manuscript could be published in The Cryosphere journal after addressing the following technical question.

- In table 1, material properties, I noticed a critical discrepancy that had caused some confusion in my first review. Material parameters used in the mathematical model seem inconsistent. The Young's modulus is for ice+permafrost+forzen roc, whereas the thermal expansion ratio is for ice/water. How do you explain this discrepancy in the material parameters? What about Poisson's ratio?

Thanks,

---

## Author Response (AR2)

The comments from the editor are shown in blue text and our responses are provided underneath in black.

Based on my own reading of the manuscript, I also feel that the corresponding descriptions of the main text (Section 3.3) might be unclear: i.e., the thermal stress model explanations do not explicitly state what was the material model or its geometry. The temperature profile is considered down to 15 m, which covers the active layer and the permafrost (thus, it is written in Line 240: "we model the frozen soil"). Before that, it has been mentioned that there is a sediment crust on top of sand-/siltstone bedrock (Line 125).

We have clarified this so that it is clearer that we ignore ground layering and apply the simplified representation of the ground as a homogeneous viscoelastic solid. From the available descriptions of Janssonhaugen and the Ullaberget Member sandstone we assume that the elastic properties of the bedrock may not be so different to the regolith, at least when the ground is frozen solid during the winter. Lacking detailed mechanical test data that could have provided the additional data needed to constrain a layered ground model, it therefore seemed most reasonable to simply assume a homogeneous solid (containing planes of weakness in the form of ice veins/wedges). The Young's modulus is in the range we might expect for either frozen regolith or soft sandstone and we assume it depends primarily on the degree of ice saturation, which increases with increasingly negative temperature. An additional point of the assumed homogeneous viscoelastic solid is that we can speculate that cryoturbation processes may play an important role in forming the regolith layer, whose depth corresponds approximately to the peak modelled thermal stress. If we assumed a softer regolith crust overlying bedrock we would enhance the thermal stress peak at the interface depth, but would lose the ability to connect that the depth of the regolith may be explained by cryoturbation processes.

However, the reader is informed about the viscoelastic rheology without learning how the heterogeneous double-layer structure was simplified in the first place (or without clarity about what were the materials of such frozen soil, which is apparently a composite material). This is particularly confusing since Table 1 is a mix of parameters corresponding (primarily) to non-composite materials, especially as it concerns the viscous parameters. Please clarify these aspects, perhaps, by mentioning how your rheological model relates to Fig.1.

This is a good point and we have clarified that we imposed the simplification of assuming a homogeneous viscoelastic solid containing planes of weakness in the form of ice veins/wedges. Lacking detailed mechanical test data from the bedrock and regolith at the study site, we think that accounting for the double-layer structure, while entirely possible, would simply add an additional degree of uncertainty without adding significantly to our ability to explain the seismic observations.

In addition, I am not sure about the logic flow in Section 3.3.1. In the beginning, the fracture of frozen soil is discussed but a reference is immediately made to the tensile strength of pure ice (Table 1). Only at the end, it is added that actually, the aim is to model ice cracking, i.e., not of frozen soil, which are two different materials.

You're right, we found that the logic flow could be improved by changing the structuring of the paragraph so that we start with the statement of what we seek to model.

Finally, please also consider the following technical corrections from my side.

Thanks for these, we have made all changes suggested.